nature
human behaviour
# How young children integrate information sources to infer the meaning of words

Manuel Bohn [1,2,4 ✉], Michael Henry Tessler [3,4], Megan Merrick[2] and Michael C. Frank[2]

**Before formal education begins, children typically acquire a vocabulary of thousands of words. This learning process requires the use of many different information sources in their social environment, including their current state of knowledge and the context in which they hear words used. How is this information integrated? We specify a developmental model according to which children consider information sources in an age-specific way and integrate them via Bayesian inference. This model accurately predicted 2–5-year-old children's word learning across a range of experimental conditions in which they had to integrate three information sources. Model comparison suggests that the central locus of development is an increased sensitivity to individual information sources, rather than changes in integration ability. This work presents a developmental theory of information integration during language learning and illustrates how formal models can be used to make a quantitative test of the predictive and explanatory power of competing theories.**

Human communicative abilities are unrivalled in the animal kingdom[1–3]. Language—in whatever modality—is the medium that allows humans to collaborate and coordinate in species-unique ways, making it the bedrock of human culture and society[4]. Thus, to absorb the culture around them and become functioning members of society, children need to learn language[5]. A central problem in language learning is referent identification: to acquire the conventional symbolic relation between a word and an object, a child must determine the intended referent of the word. There is no unique cue to reference, however, that can be used across all situations[6]. Instead, referents can only be identified inferentially by reasoning about the speaker's intentions[7–10]. That is, the child has to infer what the speaker is communicating about on the basis of information sources in the utterance's social context.

From early in development, children use several different mechanisms to harness social-contextual information sources[7,9,11]. Children expect speakers to use new words for unknown objects[12–15], to talk about objects that are relevant[16,17], new in context[18,19] or related to the ongoing conversation[20–22]. These different mechanisms, however, have been mainly described and theorized about in isolation. The implied picture of the learning process is that of a 'bag of tricks': mechanisms that operate (and develop) independently from one another[11]. As such, this view of the learning process does not address the complexity of natural social interaction during which many sources of information are present[6,23]. How do children arbitrate between these sources to accurately infer a speaker's intention?

When information integration is studied directly, the focus is mostly on how children interpret or learn words in light of social-contextual information[24–32]. In one classic study[33], children faced a four-compartment (2×2) shelf with a ball, a pen and two glasses in it. The speaker, sitting on the opposite side of the display, saw only three of the four compartments: the ball, the pen and one of the glasses. When the speaker asked for "the glass", children had to integrate the semantics of the utterance with the speaker's visual perspective to correctly infer which of the glasses the speaker was referring to. This study advanced our understanding by

documenting that preschoolers use both information sources, a finding confirmed by a variety of other work[26,29,31]. Yet these studies neither specify nor test the process by which children integrate different information sources. When interpreting such findings, work in this tradition refers to social-pragmatic theories of language use and learning[9,10,34–36], all of which assume that information is integrated as part of a social inference process but none of which clearly defines the process. As a consequence, we have no explicit and quantitative theory of how different information sources (and word-learning mechanisms) are integrated.

We present a theory of this integration process. Following social-pragmatic theories of language learning[9,10], our theory is based on the following premises: information sources serve different functional roles but are combined as part of an integrated social inference process[34–37]. Children use all available information to make inferences about the intentions behind a speaker's utterance, which then leads them to correctly identify referents in the world and learn conventional word–object mappings. We formalize the computational steps that underlie this inference process in a cognitive model[38–40]. In contrast to earlier modelling work, we treat word learning as the outcome of a social inference process and not just a cross-situational[41,42] or principle-based learning process[43]. In the remainder of this paper, we rigorously test this theory by asking how well it serves the two purposes of any psychological theory: prediction and explanation[44,45]. First, we use the model to make quantitative predictions about children's behaviour in new situations— predictions we test against new data. This form of model testing has been successfully used with adults[38,46] and here we extend it to children. Next, we quantify how well the model explains the integration process by comparing it to alternative models that make different assumptions about whether information is integrated, how it is integrated and how the integration process develops. Alternative models either assume that children ignore some information sources or—in line with a 'bag of tricks' approach—assume that children compute isolated inferences and then weigh their outcome in a posthoc manner.

[1]Department of Comparative Cultural Psychology, Max Planck Institute for Evolutionary Anthropology, Leipzig, Germany. [2]Department of Psychology, Stanford University, Stanford, CA, USA. [3]Department of Brain and Cognitive Sciences, Massachusetts Institute of Technology, Cambridge, MA, USA. [4]These authors contributed equally: Manuel Bohn, Michael Henry Tessler. ✉e-mail: manuel_bohn@eva.mpg.de

We focus on three information sources that play a central part in theorizing about language use and learning: (1) expectations that speakers communicate in a cooperative and informative manner[12,16,35], (2) shared common ground about what is being talked about in conversation[36,47,48] and (3) semantic knowledge about previously learned word–object mappings[11,49].

Our rational-integration model arbitrates between information sources via Bayesian inference (Fig. 1f gives model formulae). A listener ($L_1$) reasons about the referent of a speaker's ($S_1$) utterance. This reasoning is contextualized by the prior probability $\rho$ of each referent. We treat $\rho$ as a conversational prior which originates from the common ground shared between the listener and the speaker. This interpretation follows from the social nature of our experiments (below). From a modelling perspective, $\rho$ can be (and in fact has been) used to capture non-social aspects of a referent, for example its visual salience[38]. To decide between referents, the listener ($L_1$) reasons about what a rational speaker ($S_1$) with informativeness $\alpha$ would say given an intended referent. This speaker is assumed to compute the informativity for each available utterance and then choose the most informative one. The informativity of each utterance is given by imagining which referent a listener, who interprets words according to their literal semantics (what we call a literal listener, $L_0$), would infer on hearing the utterance. Naturally, this reasoning depends on what kind of semantic knowledge $\theta_j$ (for object $j$) the speaker ascribes to the (literal) listener.

Taken together, this model provides a quantitative theory of information integration during language learning. The three information sources operate on different timescales: speaker informativeness is a momentary expectation about a particular utterance, common ground grows over the course of a conversation and semantic knowledge is learned across development. This interplay of timescales has been hypothesized to be an important component of word meaning inference[42,50] and we link these different time-dependent processes together via their proposed impact on model components. Furthermore, the model presents an explicit and substantive theory of development. It assumes that, while children's sensitivity to the individual information sources increases with age, the way integration proceeds remains constant[7,51]. In the model, this is accomplished by creating age-dependent parameters capturing developmental changes in sensitivity to speaker informativeness ($\alpha_i$, Fig. 1d), the common ground ($\rho_i$, Fig. 1c) and object-specific semantic knowledge ($\theta_{ij}$, Fig. 1e).

To test the predictive and explanatory power of our model, we designed a word-learning experiment in which we jointly manipulated the three information sources (Fig. 1). Children interacted on a tablet computer with a series of storybook speakers[52]. This situation is depicted in Fig. 1a(4), in which a speaker (here, a frog) appears with a known object (a duck, right) and an unfamiliar object (the diamond-shaped object, left). The speakers used a new word (for example, "wug") in the context of two potential referents and then the child was asked to identify a new instance of the new word, testing their inference about the speaker's intended referent. To vary the strength of the child's inference, we systematically manipulated the familiarity of the known object (from, for example, the highly familiar "duck" to the relatively unfamiliar "pawn") and whether the familiar or new object was new to the speaker (that is, whether it was part of common ground).

This paradigm allows us to examine the integration of the three information sources described above. First, the child may infer that a cooperative and informative speaker[12,16] would have used the word "duck" to refer to the known object (the duck); the fact that the speaker did not say "duck" then suggests that the speaker is most likely referring to a different object (the unfamiliar object). This inference is often referred to as a 'mutual exclusivity' inference[13,15]. Second, the child may draw on what has already been established in the common ground with the speaker. Listeners expect speakers to

communicate about things that are new to the common ground[18,19]. Thus, the inference about the new word referring to the unfamiliar object also depends on which object is new in context (Fig. 1a,b(1)–(3)). Finally, the child may use their previously acquired semantic knowledge; that is, how sure they are that the known object is called "duck". If the known object is something less familiar, such as a chess piece (for example, a pawn), a 3-year-old child may draw a weaker inference, if they draw any inference at all[53–55]. Taken together, the child has the opportunity to integrate their assumptions about (1) cooperative communication, (2) their understanding of the common ground and (3) their existing semantic knowledge. In one condition of the experiment, information sources were aligned (Fig. 1a) while in the other they were in conflict (Fig. 1b).

## Results

**Predicting information integration across development.** We tested the model in its ability to predict 2–5-year-old children's judgments about word meaning. We estimated children's ($n = 148$) developing sensitivities to individual information sources in two separate experiments (Experiments 1 and 2; Fig. 1c-e). In Experiment 1, we estimated children's sensitivity to informativeness jointly with their semantic knowledge. In Experiment 2, we estimated sensitivity to common ground. We then generated parameter-free a priori model predictions (developmental trajectories) representing the model's expectations about how children should behave in new situations in which all three information sources had to be integrated. We generated predictions for 24 experimental conditions: 12 objects of different familiarities (requiring different levels of semantic knowledge), with novelty either conflicting or coinciding (Fig. 1). We compared these predictions to newly collected data from $n = 220$ children from the same age range (Experiment 3). All procedures, sample sizes and analyses were preregistered (Methods).

The results showed a very close alignment between model predictions and the data across the entire age range. That is, the average developmental trajectories predicted by the model resembled the trajectories found in the data (Supplementary Fig. 6). With predictions and data binned by child age (in years), the model explained 79% of the variance in the data (Fig. 2a). These results support the assumption of the model that children integrate all three available information sources.

It is still possible, however, that simpler models might make equally good—or even better—predictions. For example, work on children's use of statistical information during morphology learning showed that children's behaviour was best explained by a model that selectively ignored parts of the input[56]. Thus, we formalized the alternative view that children selectively ignore information sources in the form of three lesioned models (Fig. 2b). These models assume that children follow the heuristic 'ignore $x$' (with $x$ being one of the information sources) when multiple information sources are presented together.

The no-word-knowledge model uses the same model architecture as the rational-integration model. It uses expectations about speaker informativeness and common ground but omits semantic knowledge that is specific to the familiar objects (that is, uses only general semantic knowledge). The model assumes a listener whose inferences do not vary depending on the particular familiar object but only on the age-specific average semantic knowledge (a marker of gross vocabulary size). The no-common-ground model takes in object-specific semantic knowledge and speaker informativeness but ignores common ground information. Instead of assuming that one object has a higher prior probability to be the referent because it is new in context, the listener thinks that both objects are equally likely to be the referent. As a consequence, the listener does not differentiate between situations in which common ground is aligned or in conflict with the other information sources. Finally, according to the no-speaker-informativeness model, the listener does not

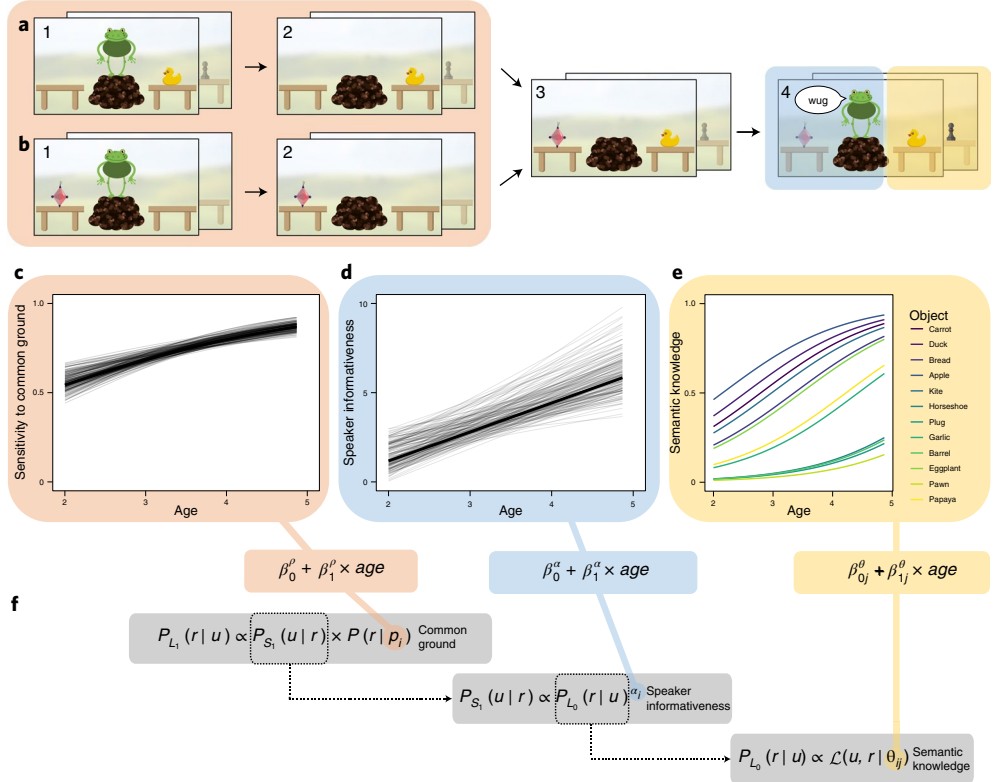

**Fig. 1 | Experimental task and model. a,b,** Screenshots from the experimental task showing the condition of the experiment in which common ground information is congruent (that is, points to the same object) with speaker informativeness (**a**) and also showing the incongruent condition (**b**). The congruent and incongruent conditions are each paired with the 12 known objects, resulting in 24 unique conditions. Steps shown are: the speaker encounters one object and then leaves the scene (1), while the speaker is away (2), a second object appears (3), when returning, the speaker uses a new word to request an object (4). Steps (1) to (3) establish common ground between the speaker and the listener, in that one object is new in context (red). The request in (4) licences an inference based on expectations about how informative speakers are (blue). Listeners' semantic knowledge enters the task because the identity of the known object on one of the tables is varied from well-known objects such as a duck to relatively unfamiliar objects such as a chess pawn (total of 12 objects, yellow). **c–e,** Developmental trajectories are shown for sensitivity to common ground (**c**), speaker informativeness (**d**) and semantic knowledge (**e**), estimated on the basis of Experiments 1 and 2 (main text). **f,** The model equation for the rational-integration model, linking information sources to model parameters.

assume that the speaker is communicating in an informative way and hence ignores the utterance. As a consequence, the inference is solely based on common ground expectations.

We found little support for these heuristic models (Fig. 2b). When using Bayesian model comparison via marginal likelihood of the data[57], the data were several orders of magnitude more likely under the rational-integration model compared to any of the lesioned models (rational integration versus no word knowledge: $BF_{10} = 3.9 \times 10^{35}$; rational integration versus no common ground: $BF_{10} = 2.6 \times 10^{47}$; rational integration versus no speaker informativeness: $BF_{10} = 4.8 \times 10^{110}$; Fig. 2). Figure 2c exemplifies the differences between the models: all heuristic models systematically underestimated children's performance in the congruent condition. Thus, even when the information sources were redundant (that is, they all point to the same referent), children's inferences were notably strengthened by each of them. In the incongruent condition, the no-word-knowledge model underestimated performance because it did not differentiate between the different familiar objects. In the case of a highly familiar word such as "duck", it therefore underestimated the effect of the utterance. The no-speaker-informativeness model completely ignored semantic knowledge, which led to even worse predictions. In contrast to the lesioned models that underestimated performance, the no-common-ground model overestimated performance in the incongruent condition because it ignored the dampening effect of common ground favouring the familiar object

as the referent. Taken together, we conclude that children considered all available information sources.

**Explaining the process of information integration.** In the previous section, we established that children integrated all available information sources to infer the meanings of new words. This result, however, does not speak to the process by which information is assumed to be integrated. Thus, in this section, we ask which integration process best explains children's behaviour.

The rational-integration model assumes that all information sources enter into a joint inference process but alternative integration processes are conceivable and might be consistent with the data. For example, the 'bag of tricks'[11] idea mentioned in the introduction could be restated as a modular integration process: children might compute independent inferences on the basis of subsets of the available information and then integrate them in a posthoc manner by weighting them according to some parameter. This view would allow for the possibility that some information sources are considered to be more important than others. In other words, children might be biased towards some information sources. We formalized this alternative view as a biased-integration model. This model assumes that semantic knowledge and expectations about speaker informativeness enter into one inference (mutual exclusivity inference)[12,13,53] while common ground information enters into a second inference. The outcomes of both processes are then weighted according to a

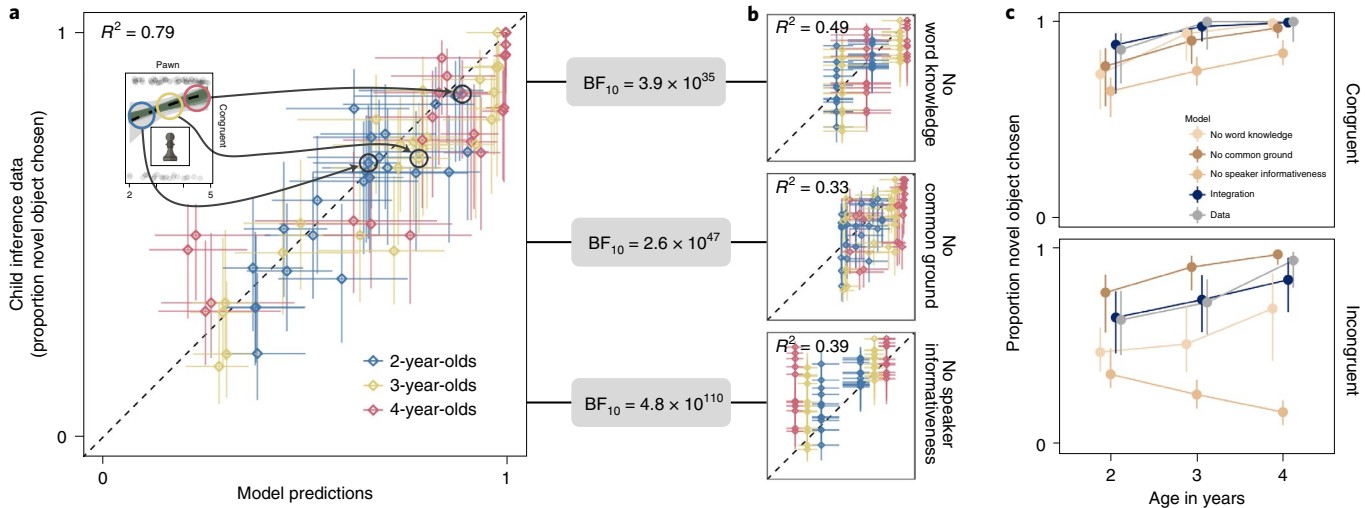

**Fig. 2 | Predicting information integration. a,b,** Correlation between model predictions and child inference data for all 24 conditions and for each age group (binned by year) for the rational-integration model (**a**) and the three lesioned models (**b**). Horizontal and vertical error bars show 95% HDI. Inset in **a** shows an example of model predictions as developmental trajectories (Fig. 3). $BF_{10}$ gives the Bayes factor in favour of the integration model based on the marginal likelihood of the data under each model. **c,** Predictions from all models considered alongside the data (with 95% HDI) for two experimental conditions (familiar word: "duck").

bias parameter $\phi$. Like the rational-integration model, this model takes in all available information sources in an age-sensitive way and assumes that they are integrated. The only difference lies in the nature of the integration process: the biased-integration model privileges some information sources over others in an ad-hoc manner.

The parameter $\phi$ in the biased-integration model is unknown ahead of time and has to be estimated on the basis of the experimental data. That is, through Experiments 1 and 2 alone, we do not learn anything about the relative importance of the information sources. As a consequence—and in contrast to the rational-integration model—the biased-integration model does not allow us to make a-priori predictions about the new data (Experiment 3) in the way we described above. For a fair comparison, we therefore constrained the parameters in the rational-integration model with the data from Experiment 3 as well. As a consequence, both models estimated their parameters using all the data available in a fully Bayesian manner (Supplementary Fig. 4).

The biased-integration model made reasonable posterior predictions and explained 78% of the variance in the data (Fig. 3b). The parameter $\phi$—indicating the bias to one of the inferences—was estimated to favour the mutual exclusivity inference (maximum a-posteriori estimate = 0.65; 95% highest density interval (HDI): 0.60–0.71; Fig. 3d). However, the rational-integration model presented a much better fit to the data, both in terms of correlation and the marginal likelihood of the data (Bayes Factor in favour of the rational-integration model: $BF_{10} = 2.1 \times 10^8$; Fig. 3b). When constrained by the data from all experiments, the rational-integration model explained 87% of the variance in the data. Figure 3e exemplifies the difference between the models: the biased-integration model put extra weight on the mutual exclusivity inference and thus failed to capture performance when this inference was weak compared to the common ground inference—such as in the congruent condition for younger children. As a result, a fully integrated—as opposed to a modular and biased—integration process explained the data better.

The rational-integration model assumes that the integration process itself does not change with age[7]. That is, while children's sensitivity to each information source develops, the way the information sources relate to one another remains the same. The biased-integration model can provide the basis for an alternative

proposal about developmental change, one in which the integration process itself changes with age. That is, children may be biased towards some information sources and that bias itself may change with age. We formalize such an alternative view as a developmental-bias model which is structurally identical to the biased-integration model but in which the parameter $\phi$ changes with age. The model assumes that the importance of the different information sources changes with age.

The developmental-bias model also explained a substantial portion of the variance in the data: 78% (Fig. 3c). The estimated developmental trajectory for the bias parameter $\phi$ suggests that younger children put a stronger emphasis on common ground information, while older children relied more on the mutual exclusivity inference (Fig. 3d). The relative importance of the two inferences seemed to switch at around age 3 yr. Yet again, when we directly compared the competitor models, we found that the data were several orders of magnitude more likely under the rational-integration model (Bayes Factor in favour of the rational-integration model: $BF_{10} = 1.4 \times 10^6$; Fig. 3b). Looking at Fig. 3e, we can see that the developmental-bias model tended to underestimate children's performance because the supportive interplay between the different inferences is constrained. In the biased models, the overall inference could only be as strong as the strongest of the components—in the rational-integration model, the components interacted with one another, allowing for stronger inferences than the individual parts would suggest.

## Discussion

The environment in which children learn language is complex. Children have to integrate different information sources, some of which relate to expectations in the moment, others to the dynamics of the unfolding interactions and yet others to their previously acquired knowledge. Our findings show that young children can integrate multiple information sources during language learning—even from relatively early in development. To answer the question of how they do so, we presented a formal cognitive model that assumes that information sources are rationally integrated via Bayesian inference.

Previous work on the study of information integration during language comprehension focused on how adults combine perceptual, semantic or syntactic information[58–62]. Our work extends this work to the development of pragmatics. Our model is based

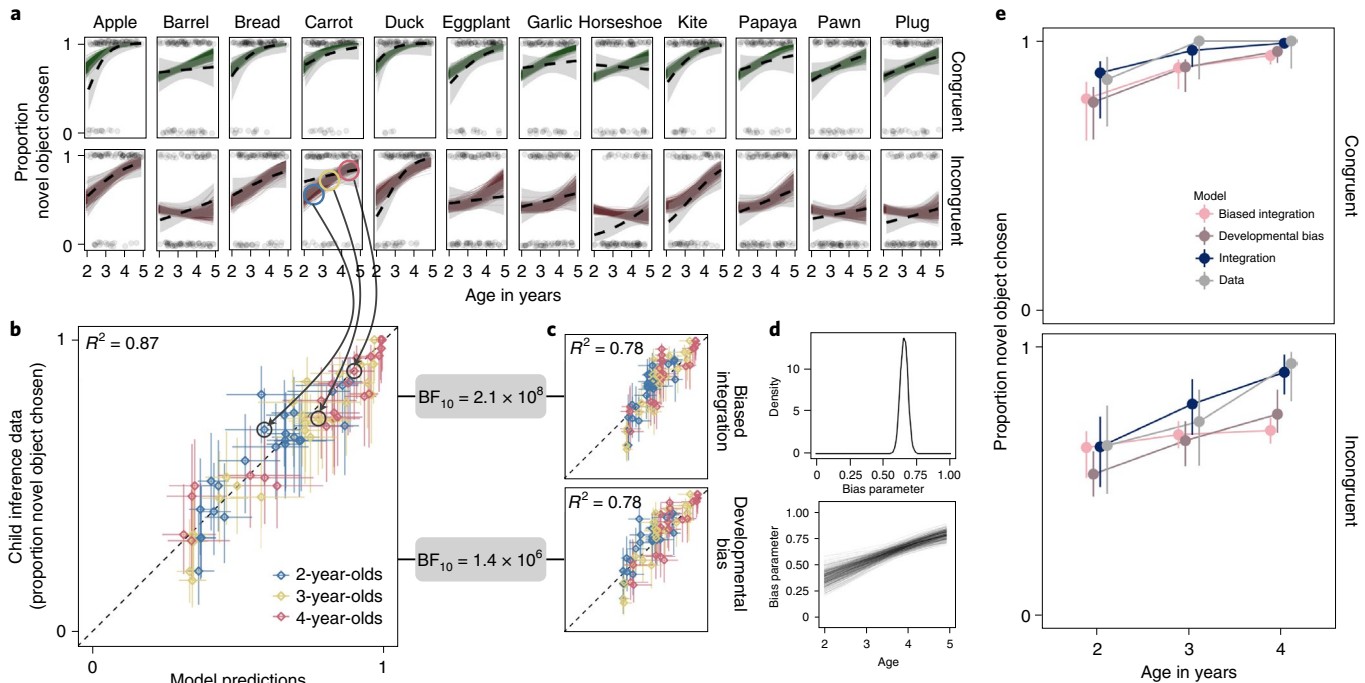

**Fig. 3 | Explaining information integration. a**, Model predictions from the rational-integration model (coloured lines) next to the behavioural data (dotted black lines with 95% CI in grey) for all 24 experimental conditions. Top row shows the congruent condition, while bottom row shows the incongruent condition. Familiar objects are ordered on the basis of their rated age of acquisition (left to right). Light dots represent individual data points. **b,c**, Correlations between model predictions binned by age and condition for the integration model (**b**) and the two biased models (**c**). Vertical and horizontal error bars show 95% HDIs. $BF_{10}$ gives the Bayes factor in favour of the rational-integration model on the basis of the marginal likelihood of the data under each model. **d**, Posterior distribution of the bias parameter in the biased-integration model and developmental trajectories for the bias parameter in the developmental-bias model. **e**, Predictions from all models considered alongside the data (with 95% HDI) for two experimental conditions (familiar word: "duck").

on classic social-pragmatic theories on language use and comprehension[10,34–36]. As a consequence, instead of assuming that different information sources feed into separate word-learning mechanisms (the 'bag of tricks' view), we assume that all of these information sources play a functional role in an integrated social inference process. Our model goes beyond previous theoretical and empirical work by describing the computations that underlie this inference process. Furthermore, we presented a substantive theory about how this integration process develops: we assume that children become increasingly sensitive to different information sources but that the way these information sources are integrated remains the same. We used this model to predict and explain children's information integration in a new word-learning paradigm in which they had to integrate (1) their assumptions about informative communication, (2) their understanding of the common ground and (3) their existing semantic knowledge.

The rational-integration model made accurate quantitative predictions across a range of experimental conditions both when information sources were aligned and when they were in conflict. Predictions from the model better explained the data compared to lesioned models which assumed that children ignore one of the information sources, suggesting that children used all available information. We also formalized an alternative, modular, view. According to the biased-integration model, children use all available information sources but compute separate inferences on the basis of a subset of them. Integration happens by weighing the outcomes of these separate inferences by some parameter. Finally, we tested an alternative view on the development of the integration process. According to the developmental-bias model, the importance of the different information sources changes with age. In both cases, the rational-integration model provided a much better fit to the data,

suggesting that the integration process remains stable over time. That is, there is developmental continuity and therefore no qualitative difference in how a 2-year-old integrates information compared to a 4-year-old.

The rational-integration model is derived from a more general framework for pragmatic inference, which has been used to explain a wide variety of phenomena in adults' language use and comprehension[38,39,63–67]. Thus, it can be generalized in a natural way to capture word learning in contexts that offer more, fewer or different types of information. For example, non-verbal aspects of the utterance (such as eye-gaze or gestures) can affect children's mutual exclusivity inferences[68–72]. As a first step in this direction, we recently studied how adults and children integrate non-verbal utterances with common ground[51]. Using a structurally similar rational-integration model, we also found a close alignment between model predictions and the data. The flexibility of this modelling framework stems from its conceptualization of human communication as a form of rational social action. As such, it connects to computational and empirical work that tries to explain social reasoning by assuming that humans expect each other to behave in a way that maximizes the benefits and minimizes the cost associated with actions[28,73,74].

Our model and empirical paradigm provide a foundation on which to test deeper questions about language development. First, our experiments should be performed in children from different cultural backgrounds learning different languages[75]. In such studies, we would not expect our results to replicate in a strict sense; that is, we would not expect to see the same developmental trajectories in all cultures and languages. Substantial variation is much more likely. Studies on children's pragmatic inferences in different cultures have documented both similar[76,77] and different[78] developmental trajectories. Nevertheless, our model provides a way to think about how

to reconcile cross-cultural variation with a shared cognitive architecture: we predict differences in how sensitive children are to the individual information sources at different ages but similarities in how information is integrated[7]. In computational terms, we assume a universal architecture that specifies the relation between a set of varying parameters. Of course, either confirmation or disconfirmation of this prediction would be informative.

Second, it would be useful to flesh out the cognitive processes that underlie reasoning about common ground. The basic assumption that common ground changes interlocutors' expectations about what are likely referents[79] has been used in earlier modelling work on the role of common ground in reference resolution[62]. Here, we went one step further and measured the strength of these expectations to inform the parameter values in our model. However, in its current form, our model treats common ground as a conversational prior and does not specify how the listener arrives at the expectation that some objects are more likely to be referents because they are new in common ground. That is, computationally, our model does not differentiate between common ground information and other reasons that might make an object contextually more salient. An interesting starting point to overcome this shortcoming would be modelling work on the role of common ground in conversational turn-taking[80].

Finally, our model is a model of referent identification in the moment of the utterance. At the same time, the constructs made use of by our model are shaped by factors that unfold across multiple time points and contexts: common ground is built over the course of a conversation and the lexical knowledge of a child is shaped across a language developmental timescale. Even speaker informativeness could be imagined to vary over time following repeated interactions with a particular speaker. What is more, assessing speaker informativeness is unlikely to be the outcome of a single, easy-to-define process. The expectations about informative communication that we take it to represent are probably the result of the interplay between multiple social and non-social inference processes. The broader point here is that our model makes use of unidimensional representations of high-dimensional, structured processes and examines how these representations are integrated. As such, it is first and foremost a computational description of the inferences and we therefore make no strong claims about the psychological reality of the parameters in it. Connecting our model with other frameworks that focus on the cognitive, temporal and cross-situational aspects of word learning would elucidate further these complex processes[42,50,81].

This work advances our understanding of how children navigate the complexity of their learning environment. Methodologically, it illustrates how computational models can be used to test theories; from a theoretical perspective, it adds to broader frameworks that see the ontogenetic and phylogenetic emergence of language as deeply rooted in social cognition.

## Methods

A more detailed description of the experiments and the models can be found in the Supplementary Information. The experimental procedure, sample sizes and analysis for each experiment were preregistered (https://osf.io/7rg9j/registrations; dates of registration: 2 May 2019, 5 April 2019 and 2 March 2019). Experimental procedures, data, model and analysis scripts can be found in an online repository (https://github.com/manuelbohn/spin). Experiments 1 and 2 were designed to estimate children's developing sensitivity to each information source. The results of these experiments determine the parameter values in the model (Fig. 1c–f). Experiment 3 was designed to test how children integrate different information sources.

**Participants.** Sample sizes for each experiment were chosen to have at least 30 data points per cell (that is, unique combination of condition, familiar object and age group). Across the three experiments, a total of 368 children participated. Experiment 1 involved 90 children, including 30 2-year-olds (range = 2.03–3.00, 15 girls), 30 3-year-olds (range = 3.03–3.97, 22 girls) and 30 4-year-olds (range = 4.03–4.90, 16 girls). Data from ten additional children were not included because they were either exposed to less than 75% of English at home (five children), did not

finish at least half of the test trials (two children), the technical equipment failed (two children) or their parents reported an autism spectrum disorder (one child).

In Experiment 2, we tested 58 children, including 18 2-year-olds (range = 2.02–2.93, seven girls), 19 3-year-olds (range = 3.01–3.90, 14 girls) and 21 4-year-olds (range = 4.07–4.93, 14 girls). Data from five additional children were not included because they were either exposed to less than 75% of English at home (three children) or the technical equipment failed (two children).

Finally, Experiment 3 involved 220 children, including 76 2-year-olds (range = 2.04–2.99, seven girls), 72 3-year-olds (range = 3.00–3.98, 14 girls) and 72 4-year-olds (range = 4.00–4.94, 14 girls). Data from 20 additional children were not included because they were either exposed to less than 75% of English at home (15 children), did not finish at least half of the test trials (three children) or the technical equipment failed (two children).

All children were recruited in a children's museum in San José, California, United States. This population is characterized by a diverse ethnic background (predominantly White, Asian or mixed-ethnicity) and high levels of parental education and socioeconomic status. Parents consented to their children's participation and provided demographic information. All experiments were approved by the Stanford Institutional Review Board (protocol no. 19960).

**Materials.** All experiments were presented as an interactive picture book on a tablet computer. Tablet-based storybooks are commonly used to simulate social interactions in developmental research and interventions[82]. A recent, direct comparison found similar performance with tablet-based and printed storybooks in a word-learning paradigm[52]. Furthermore, our results in Experiment 1 and 2 replicate earlier studies on mutual exclusivity and discourse novelty that used live interactions instead of storybooks[18,19].

Figure 1a,b show screenshots from the actual experiments. The general setup involved an animal standing on a little hill between two tables. For each animal character, we recorded a set of utterances (one native English speaker per animal) that were used to talk to the child and make requests. Each experiment started with two training trials in which the speaker requested known objects (car and ball).

**Procedure.** Experiment 1 tested the mutual exclusivity inference[13,53]. On one table, there was a familiar object; on the other table, there was an unfamiliar object (a new design drawn for the purpose of the study) (Fig. 1a/b(4) and Supplementary Fig. 1a). The speaker requested an object by saying "Oh cool, there is a (non-word) on the table, how neat, can you give me the (non-word)?". Children responded by touching one of the objects. The location of the unfamiliar object (left or right table) and the animal character were counterbalanced. We coded a response as a correct choice if children chose the unfamiliar object as the referent of the new word. Each child completed 12 trials, each with a different familiar and a different unfamiliar object. We used familiar objects that we expected to vary along the dimension of how likely children were to know the word for it. This set included objects that most 2-year-olds can name (for example, a duck) as well as objects that only very few 5-year-olds can name (for example, a pawn (chess piece)). The selection was based on the age of acquisition ratings from Kuperman and colleagues[83]. While these ratings usually do not capture the absolute age when children acquire these words, they capture the relative order in which words are learned. Supplementary Fig. 2a shows the words and objects used in the experiment. There was a high correlation between the rated age-of-acquisition and the mutual exclusivity effect for the different words (Supplementary Fig. 2c).

Experiment 2 tested children's sensitivity to common ground that is built up over the course of a conversation. In particular, we tested whether children keep track of which object is new to a speaker and which they have encountered previously[18,19]. The general setup was the same as in Experiment 1 (Supplementary Fig. 1b). The speaker was positioned between the tables. There was an unfamiliar object (drawn for the purpose of the study) on one of the tables while the other table was empty. Next, the speaker turned to one of the tables and either commented on the presence ("Aha, look at that.") or the absence ("Hm, nothing there.") of an object. Then the speaker disappeared. While the speaker was away, a second unfamiliar object appeared on the previously empty table. Then the speaker returned and requested an object in the same way as in Experiment 1. The positioning of the unfamiliar object at the beginning of the experiment, the speaker as well as the location the speaker turned to first was counterbalanced. Children completed ten trials, each with a different pair of unfamiliar objects. We coded a response as a correct choice if children chose as the referent of the new word the object that was new to the speaker.

Experiment 3 combined the procedures from Experiments 1 and 2. It followed the same procedure as Experiment 2 but involved the same objects as Experiment 1 (Fig. 1(1)–(4)) and Supplementary Fig. 1c. In the beginning, one table was empty while there was an object (unfamiliar or familiar) on the other one. After commenting on the presence or absence of an object on each table, the speaker disappeared and a second object appeared (familiar or unfamiliar). Next, the speaker reappeared and made the usual request ("Oh cool, there is a (non-word) on the table, how neat, can you give me the (non-word)?"). In the congruent condition, the familiar object was present in the beginning and the unfamiliar object appeared while the speaker was away (Fig. 1a and Supplementary Fig. 1c,

left). In this case, both the mutual exclusivity and the common ground inference pointed to the new object as the referent (that is, it was both new to the speaker in the context and it was an object that does not have a label in the lexicon). In the incongruent condition, the unfamiliar object was present in the beginning and the familiar object appeared later. In this case, the two inferences pointed to different objects (Fig. 1b and Supplementary Fig. 1c, right). This resulted in a total of two alignments (congruent versus incongruent) × 12 familiar objects = 24 different conditions. Participants received up to 12 test trials, six in each alignment condition, each with a different familiar and unfamiliar object. Familiar objects were the same as in Experiment 1. The positioning of the objects on the tables, the speaker and the location the speaker first turned to were counterbalanced. Participants could stop the experiment after six trials (three per alignment condition). If a participant stopped after half of the trials, we tested an additional participant from the same age group to reach the preregistered number of data points per age group (2-, 3- and 4-year-olds).

**Data analysis.** To analyse how the manipulations in each experiment affected children's behaviour, we used generalized linear mixed models. Since the focus of the paper is on how information sources were integrated, we discuss these models in the Supplementary Information and focus here on the cognitive models instead. A detailed, mathematical description of the different cognitive models along with details about estimation procedures and priors can be found in the Supplementary Information. All cognitive models and Bayesian data analytic models were implemented in the probabilistic programming language WebPPL[84]. The corresponding model code can be found in the associated online repository. Information about priors for parameter estimation and Markov chain Monte Carlo settings can also be found in the Supplementary Information and the online repository.

As a first step, we used the data from Experiments 1 and 2 to estimate children's developing sensitivity to each information source. To estimate the parameters for semantic knowledge ($\theta$) and speaker informativeness ($\alpha$), we adapted the rational-integration model to model a situation in which both objects (new and familiar) have equal prior probability (that is, no common ground information). We used the data from Experiment 1 to then infer the semantic knowledge and speaker informativeness parameters in an age-sensitive manner. Specifically, we inferred the intercepts and slopes for speaker informativeness via a linear regression submodel and semantic knowledge via a logistic regression submodel, the values of which were then combined in the cognitive model to generate model predictions to predict the responses generated in Experiment 1. To estimate the parameters representing sensitivity to common ground ($\rho$), we used a simple logistic regression to infer which combination of intercept and slope would generate predictions that corresponded to the average proportion of correct responses measured in Experiment 2. For the 'prediction' models, the parameters whose values were inferred by the data from Experiments 1 and 2 were then used to make out-of-sample predictions for Experiment 3. For the 'explanation' models, these parameters were additionally constrained by the data from Experiment 3. A more detailed description of how these parameters were estimated (including a graphical model, Supplementary Fig. 4) can be found in the Supplementary Information.

To generate model predictions, we combined the parameters according to the respective model formula. As mentioned above, common ground information could either be aligned or in conflict with the other information sources. In the congruent condition, the unfamiliar object was also new in context and thus had the prior probability $\rho$. In the incongruent condition, the new object was the 'old' object and thus had the prior probability of $1 - \rho$.

The rational-integration model is a mapping from an utterance $u$ to a referent $r$, defined as

$$P_{L_1}^{int}(r \mid u; \{\rho_i, \alpha_i \theta_{ij}\}) \propto P_{S_1}(u \mid r; \{\alpha_i, \theta_{ij}\}) \times P(r \mid \rho_i)$$

where $i$ represents the age of the participant and the $j$ the familiar object. The three lesioned models that were used to compare how well the model predicts new data are reduced versions of this model. The no-word-knowledge model uses the same model architecture

$$P_{L_1}^{no\_wk}(r \mid u; \{\rho_i, \alpha_i \theta_i\}) \propto P_{S_1}(u \mid r; \{\alpha_i, \theta_i\}) \times P(r \mid \rho_i)$$

and the only difference lies in the parameter $\theta$, which does not vary as a function of $j$, the object (that is, $\theta$ in this model is analogous to a measure of gross vocabulary development). The object-specific parameters for semantic knowledge are fitted via a hierarchical regression (mixed effects) model. That is, there is an overall developmental trajectory for semantic knowledge (main effect, $\theta_i$) and then there is object-specific variation around this trajectory (random effects, $\theta_{ij}$). Thus, the no-word-knowledge model takes in the overall trajectory for semantic knowledge ($\theta_i$) but ignores object-specific variation. The no-common-ground model ignores common ground information (represented by $\rho$) and is thus defined as

$$P_{L_1}^{no\_cg}(r \mid u; \{\alpha_i \theta_{ij}\}) \propto P_{S_1}(u \mid r; \{\alpha_i, \theta_{ij}\}).$$

For the no-speaker-informativeness model, the parameter $\alpha = 0$. As a consequence, the likelihood term in the model is 1 and the model therefore reduces to

$$P_{L_1}^{no\_si}(r \mid u; \{\rho_i\}) \propto P(r \mid \rho_i).$$

As noted above, the explanation models used parameters that were additionally constrained by the data from Experiment 3 but the way these parameters were combined in the rational-integration model was the same as above. The biased-integration model is defined as

$$P_{L_1}^{biased}(r \mid u; \{\phi, \rho_i, \alpha_i, \theta_{ij}\}) = \phi \times P_{ME}(r \mid u; \{\alpha_i, \theta_{ij}\}) + (1 - \phi) \times P(r \mid \rho_i)$$

with $P_{ME}$ representing a mutual exclusivity inference which takes in speaker informativeness and object-specific semantic knowledge. This inference is then weighted by the parameter $\phi$ and added to the respective prior probability, which is weighted by $1 - \phi$. Thus, $\phi$ represents the bias in favour of the mutual exclusivity inference. In the developmental-bias model the parameter $\phi$ is made to change with age ($\phi$) and the model is thus defined as

$$P_{L_1}^{dev\_bias}(r \mid u; \{\phi_i, \rho_i, \alpha_i, \theta_{ij}\}) = \phi_i \times P_{ME}(r \mid u; \{\alpha_i, \theta_{ij}\}) + (1 - \phi_i).$$

We compared models in two ways. First, we used Pearson correlations between model predictions and the data. For this analysis, we binned the model predictions and the data by age in years and by the type of familiar object (Figs. 2 and 3 and Supplementary Figs. 7 and 10). Second, we compared models on the basis of the marginal likelihood of the data under each model—the likelihood of the data averaging over ('marginalizing over') the prior distribution on parameters; the pairwise ratio of marginal likelihoods for two models is known as the Bayes Factor. It is interpreted as how many times more likely the data are under one model compared to the other. Bayes Factors quantify the quality of predictions of a model, averaging over the possible values of the parameters of the models (weighted by the prior probabilities of those parameter values); by averaging over the prior distribution on parameters, Bayes Factors implicitly take into account model complexity because models with more parameters will tend to have a broader prior distribution over parameters, which in effect, can water down the potential gains in predictive accuracy that a model with more parameters can achieve[57]. For this analysis, we treated age continuously.

**Reporting Summary.** Further information on research design is available in the Nature Research Reporting Summary linked to this article.

## Data availability
Data files, along with all experimental stimuli, model and analysis scripts can be found at: https://github.com/manuelbohn/spin.

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

## Acknowledgements

M.B. received funding from the European Union's Horizon 2020 research and innovation programme under the Marie Sklodowska-Curie grant agreement no. 749229. M.H.T. was funded by the National Science Foundation SBE Postdoctoral Research Fellowship grant no. 1911790. M.C.F. was supported by a Jacobs Foundation Advanced Research Fellowship and the Zhou Fund for Language and Cognition. The funders had no role in study design, data collection and analysis, decision to publish, or preparation of the manuscript.

## Author contributions

M.B., M.H.T. and M.C.F. conceptualized the study. M.M. collected the data. M.B. and M.H.T. implemented the models and analysed the data. M.B., M.H.T. and M.C.F. wrote the manuscript; all authors approved the final version of the manuscript.

## Competing interests

The authors declare no competing interests.

## Additional information

**Correspondence and requests for materials** should be addressed to M.B.

# Reporting Summary

Nature Research wishes to improve the reproducibility of the work that we publish. This form provides structure for consistency and transparency in reporting. For further information on Nature Research policies, see Authors & Referees and the Editorial Policy Checklist.

## Statistics

For all statistical analyses, confirm that the following items are present in the figure legend, table legend, main text, or Methods section.

| n/a | Confirmed | |
|---|---|---|
| ☐ | ☒ | The exact sample size (*n*) for each experimental group/condition, given as a discrete number and unit of measurement |
| ☐ | ☒ | A statement on whether measurements were taken from distinct samples or whether the same sample was measured repeatedly |
| ☐ | ☒ | The statistical test(s) used AND whether they are one- or two-sided<br>*Only common tests should be described solely by name; describe more complex techniques in the Methods section.* |
| ☐ | ☒ | A description of all covariates tested |
| ☒ | ☐ | A description of any assumptions or corrections, such as tests of normality and adjustment for multiple comparisons |
| ☐ | ☒ | A full description of the statistical parameters including central tendency (e.g. means) or other basic estimates (e.g. regression coefficient) AND variation (e.g. standard deviation) or associated estimates of uncertainty (e.g. confidence intervals) |
| ☒ | ☐ | For null hypothesis testing, the test statistic (e.g. *F*, *t*, *r*) with confidence intervals, effect sizes, degrees of freedom and *P* value noted<br>*Give P values as exact values whenever suitable.* |
| ☐ | ☒ | For Bayesian analysis, information on the choice of priors and Markov chain Monte Carlo settings |
| ☐ | ☒ | For hierarchical and complex designs, identification of the appropriate level for tests and full reporting of outcomes |
| ☐ | ☒ | Estimates of effect sizes (e.g. Cohen's *d*, Pearson's *r*), indicating how they were calculated |

*Our web collection on statistics for biologists contains articles on many of the points above.*

## Software and code

Policy information about availability of computer code

| Data collection | All data was collected using custom made picture book style web-experiments (implemented in HTML/ JavaScript). Experiments (and code) can be found in the associated online respository (https://github.com/manuelbohn/spin). |
|---|---|
| Data analysis | Data processing and model comparisons were done in R.  Probabilistic models and model comparisons were implemented in WebPPL. All analysis code can be found in the supplementary material and the associated online repository (https://github.com/manuelbohn/spin). |

For manuscripts utilizing custom algorithms or software that are central to the research but not yet described in published literature, software must be made available to editors/reviewers. We strongly encourage code deposition in a community repository (e.g. GitHub). See the Nature Research guidelines for submitting code & software for further information.

## Data

Policy information about availability of data

All manuscripts must include a data availability statement. This statement should provide the following information, where applicable:

- Accession codes, unique identifiers, or web links for publicly available datasets
- A list of figures that have associated raw data
- A description of any restrictions on data availability

Experimental stimuli, data files and analysis scripts are freely available in an online repository (https://github.com/manuelbohn/spin)

# Field-specific reporting

Please select the one below that is the best fit for your research. If you are not sure, read the appropriate sections before making your selection.

☐ Life sciences    ☒ Behavioural & social sciences    ☐ Ecological, evolutionary & environmental sciences

# Behavioural & social sciences study design

All studies must disclose on these points even when the disclosure is negative.

| | |
|---|---|
| Study description | Quantitative experimental studies with cognitive modelling component |
| Research sample | Children (N = 368 , age range: 2.0 - 5.0): Visitors to the Children's Discovery Museum in San Jose, California, USA with mixed ethnic background. |
| Sampling strategy | The sample was a convenience sample. Sample sizes were set based on previous studies and pre-registered prior to data collection. |
| Data collection | Experiments were presented in the form of a website on a tablet computer. Children were guided through the experiment by an experimenter. The experimenter was blind to the purpose of the study. Two training trials, in which participants had to listen to respond to audio recordings, were placed at the beginning of each experiment to assure that participants paid attention and understood the basic procedure. Parents were present in the room but asked to sit back and fill out a demographics form. |
| Timing | Data collection for adults took place between March and August 2019. |
| Data exclusions | Participants were excluded if they responded incorrectly in 2/2 training trials in which they had to select a familiar object (car, ball) upon request. Children were further excluded if they reported to hear less than 75% of English at home or if they finished less than half of the test trials of a given experiment. |
| Non-participation | Data from 35 additional children were not included because they were either exposed to less than 75% of English at home (23), did not finish at least half of the test trials (5), the technical equipment failed (6) or their parent reported an autism spectrum disorder (1). |
| Randomization | Critical comparisons were based on conditions tested within subject. The order of conditions within subjects was randomized. Experiments were run in succession and participants were therefore assigned to whichever experiment was running at the moment. |

# Reporting for specific materials, systems and methods

We require information from authors about some types of materials, experimental systems and methods used in many studies. Here, indicate whether each material, system or method listed is relevant to your study. If you are not sure if a list item applies to your research, read the appropriate section before selecting a response.

## Materials & experimental systems

| n/a | Involved in the study |
|---|---|
| ☒ | Antibodies |
| ☒ | Eukaryotic cell lines |
| ☒ | Palaeontology |
| ☒ | Animals and other organisms |
| ☐ | ☒ Human research participants |
| ☒ | Clinical data |

## Methods

| n/a | Involved in the study |
|---|---|
| ☒ | ChIP-seq |
| ☒ | Flow cytometry |
| ☒ | MRI-based neuroimaging |

# Human research participants

Policy information about studies involving human research participants

| | |
|---|---|
| Population characteristics | See above |
| Recruitment | Parents were approached on the museum floor and asked whether they would like to participate in a short experiment on child development. Parents received no monetary compensation for their participation, children received a sticker. Due to its location, visitors to the museum constitute a WEIRD sample with exceptionally high levels of education and SES. The exact age at which children show evidence for making inferences on individual information sources might thus be different in other populations. However, we have no reason to assume that the information integration process, as specified in our model, would be different. |
| Ethics oversight | Stanford Institutional Review Board, protocol #19960. |

Note that full information on the approval of the study protocol must also be provided in the manuscript.

