## [Peer Review File · Nature Human Behaviour]

Peer Review Information

Journal: Nature Human Behaviour

Manuscript Title: How young children integrate information sources to infer the meaning of words

Corresponding author name(s): Manuel Bohn

Editorial Notes:

Reviewer Comments & Decisions:

Decision Letter, initial version:

1st February 2021

Dear Dr Bohn,

Thank you once again for your manuscript, entitled "How young children integrate information sources to infer the meaning of words", and for your patience during the peer review process. Please accept my sincerest apologies for the delay in making a decision!

Your Article has now been evaluated by 3 referees. You will see from their comments copied below that, although they find your work of potential interest, they have raised quite substantial concerns. In light of these comments, we cannot accept the manuscript for publication, but would be interested in considering a revised version if you are willing and able to fully address reviewer and editorial concerns.

We hope you will find the referees' comments useful as you decide how to proceed. If you wish to submit a substantially revised manuscript, please bear in mind that we will be reluctant to approach the referees again in the absence of major revisions. We are committed to providing a fair and constructive peer-review process. Do not hesitate to contact us if there are specific requests from the reviewers that you believe are technically impossible or unlikely to yield a meaningful outcome.

To guide the scope of the revisions, the editors discuss the referee reports in detail within the team, including with the chief editor, with a view to (1) identifying key priorities that should be addressed in revision and (2) overruling referee requests that are deemed beyond the scope of the current study. We hope that you will find the prioritised set of referee points to be useful when revising your study. Please do not hesitate to get in touch if you would like to discuss these issues further.

(A) Reviewer 1 mentions that the relevant existing literature is not appropriately integrated. When revising your manuscript, please carefully consider this comment and incorporate existing literature in

your text.

(B) Reviewer 2 and 3 point out various unclarities concerning the computational modelling part of the manuscript. Please address these comments and include additional analyses when necessary.

If you wish to submit a suitably revised manuscript we would hope to receive it within 4 months. We understand that the COVID-19 pandemic is causing significant disruptions which may prevent you from carrying out the additional work required for resubmission of your manuscript within this timeframe. If you are unable to submit your revised manuscript within 4 months, please let us know. We will be happy to extend the submission date to enable you to complete your work on the revision.

- Include a "Response to the editors and reviewers" document detailing, point-by-point, how you addressed each editor and referee comment. If no action was taken to address a point, you must provide a compelling argument. This response will be used by the editors to evaluate your revision and sent back to the reviewers along with the revised manuscript.
- Highlight all changes made to your manuscript or provide us with a version that tracks changes.

[REDACTED]

Thank you for the opportunity to review your work. Please do not hesitate to contact me if you have any questions or would like to discuss the required revisions further.

Sincerely,

Samantha Antusch, PhD
Associate editor
Nature Human Behaviour

Reviewer expertise:

Reviewer #1: language development and information integration

Reviewer #2: language development and information integration; computational models

Reviewer #3: computational modelling; child development

REVIEWER COMMENTS:

Reviewer #1:

Remarks to the Author:

In this manuscript, the authors propose a Bayesian model to account for children's integration of two different information sources in word learning, namely a "mutual exclusivity" response and attention to common ground. They test whether the model captures children's word learning performance in three different experiments.

This is an interesting paper with a laudable goal—namely, capturing children's integration of different sources of information in word learning. At the same time, I note some significant concerns with the manuscript, particularly with the lack of attention to similar models in the literature as well as the

1. The authors propose a model to account for children's integration of social-pragmatic information in word learning, drawing upon a Bayesian framework. I was quite surprised that no reference was made in the paper to Xu & Tenenbaum's 2007 model in *Psychological Review* (i.e., Word learning as Bayesian inference). I recognize that the model proposed in that paper is capturing a different aspect of word learning but it seems like it deserves some attention here.
2. In a similar vein, I found the set up for the model somewhat disconnected. That is, the model is accounting for children's integration of two particular informational sources. This is not to say that this is not important but the abstract and introduction lead the reader to expect that a wide variety of word learning conditions are tested (e.g., "a range of experimental conditions"). It would be helpful to state up front the two informational sources that are being tested.
3. Within the language processing literature, a great deal of attention has been paid to questions around integration of multiple constraints including the proposal of Bayesian models—the Heller et al model in particular (which is cited in the paper). I think it would be helpful to indicate how this proposed model connects with and extends this literature.
4. The introduction notes that studies have typically examined children's attention to information in isolation. Other theorists have sought to provide more integrative accounts of word learning (e.g., Golinkoff et al., etc...). I think some acknowledgement of this literature is relevant here. Furthermore, other studies have sought to examine children's attention to convergent and divergent cues, with particular focus on the mutual exclusivity type phenomenon that is part of the focus in this model. I note some of these papers below. Would the model account for the findings from these types of models?

Gangopadhyay, I., & Kaushanskaya, M. (2020). The role of speaker eye gaze and mutual exclusivity in novel word learning by monolingual and bilingual children. *Journal of Experimental Child Psychology*,

197, 104878.

Jaswal, V. K., & Hansen, M. B. (2006). Learning words: Children disregard some pragmatic information that conflicts with mutual exclusivity. *Developmental science*, 9(2), 158-165.

Jaswal, V. K., & Markman, E. M. (2003). The relative strengths of indirect and direct word learning. *Developmental Psychology*, 39(4), 745.

Jaswal, V. K. (2010). Explaining the disambiguation effect: Don't exclude mutual exclusivity. *Journal of Child Language*, 37(1), 95-113.

Graham, S. A., Nilsen, E. S., Collins, S., & Olineck, K. (2010). The role of gaze direction and mutual exclusivity in guiding 24 - month - olds' word mappings. *British Journal of Developmental Psychology*, 28(2), 449-465.

Paulus, M., & Fikkert, P. (2014). Conflicting social cues: Fourteen-and 24-month-old infants' reliance on gaze and pointing cues in word learning. *Journal of Cognition and Development*, 15(1), 43-59.

5. One core principle in the model is the way children integrate information sources remains constant. This is an interesting proposal and I appreciate that the model seems to capture this. I wonder, however, how this aligns with other research demonstrating increases in integrative capacity during the preschool years (i.e., changes in executive function). Can the authors expand on this proposal and align it with research demonstrating changes in EF?

6. No justification for sample sizes are provided. Notably the sample sizes in Experiment 3 are much larger than the sample sizes in previous experiments. It would be helpful to provide some justification for this decision.

Minor issues:

p. 3: It is noted that "referents can only be identified inferentially by reasoning about a speaker's intentions". This statement is not quite accurate as children can, of course, identify word referents through other means (tracking cross-situational occurrences etc...).

Line 105: word-learning experiment (not world-learning)

Reviewer #2:

Remarks to the Author:

Summary of paper

=====

The authors take on a question of pragmatics in development: How do children establish the reference of an utterance when there are multiple sources of information contributing to the inference (common ground, informativeness, prior knowledge)? The authors develop a computational model based on

their previous work with RSA, but in a developmental context, verify the parameters with two experiments, and then test it with a third, comparing it to alternative models of information integration (which do not fare as well).

I think there's a lot to like about this paper, and I think it sits well within the goals of Nature Human Behavior to develop novel cognitive models from first principles and then test them. The results will likely be of interest to people in different areas of interest across cognition such as Development, Language, Pragmatics, and Computational Modeling.

It's the nature of positive reviews that they are shorter than less-positive reviews so I don't have much to add. Please see the comments below for some suggestions, but I don't see these are barriers for publication.

Comments

=====

1. In lines 40-60, one could wrongly get the impression that the reality is that children use a "bag of tricks", and the question is now how to model it. That is, something like "Previous work has considered A, B, C in isolation but how do they work together? What're the weights a,b,c, such that $JUDGMENTS = a*A + b*B + c*C?$ ". I think you could add another few lines somewhere around line 60 to more clearly state your integration theory ISN'T a bag-of-tricks view and that you will test it against such one.
2. I understand 'sensitivity to informativeness' (alpha) is defined based on previous work but I still don't quite get from the opening how this is supposed to work. I understand a parameter relating to "do I know this word" as pretty directly grounded in some mental variable, I understand a parameter relating to common ground as "Did I notice you weren't here" (maybe), and their change over time could reasonably reflect development of competence, memory, knowledge, etc. What exactly is growing sensitivity to informativeness meant to represent? Do we think the child is able to think of the speaker as informative, but what they grow to understand is that this...matters? It seems unlikely, and probably I'm missing an obvious point here, but it seems like sensitivity to informativeness is likely tracking a bunch of more complicated processes having to do with whether the child themselves can reason about the speaker, can calculate informativeness, etc. As such, this parameter seems more like it is soaking up noise/growing competence than a specific thing like 'memory for this object'. (see also comment 5 on the interpretation of 'sensitivity to common ground').
3. Following on (2), if I understand correctly your experiments do not independently verify the parameters for informativeness and semantic knowledge, rather those are both being estimated at the same time from Exp 1. Is there a reason not to do this independently? Is it simply not possible?
4. Looking at the supp', Figure S6: What's going on with some of the cases where the prediction seems to be going in the opposite direction than the data, like horseshoe, plug, barrel, pawn? Is this just chalked up to noise or is there something more principled here?
5. (around lines 312-314) "But computationally, the model does not differentiate between common ground information and other reasons that might make an object contextually more salient..." -- Right, I had this thought earlier in the manuscript. Is there any way to tease these apart? It also makes a lot more sense to me personally to call the relevant parameter (currently 'sensitivity to

common ground') something that refers to this attention-grabbing aspect of the object, whether it is novelty, shininess, blinking, etc. Calling it 'sensitivity to common ground' seems overly specific, and like 'sensitivity to informativeness' seems unlikely to refer to a natural kind. This should probably be discussed/acknowledged much higher up when introducing the 'common ground' bits of the model.

6. Cultural differences (around lines 299-301): The authors acknowledge their results will likely not replicate in a strict sense, which is fair enough. But...what exactly do you think will be driving this cultural variation? If the underlying process is the same and the integration is the same, this leaves us with children having different knowledge of objects in different cultures (which is fair enough, perhaps in some cultures they know about more/less objects, perhaps not _these objects), or having 'different sensitivity to common ground / informativeness'. The latter is much weirder and you'd need to explain even briefly how that would be driven by a cultural difference. I'm not expecting an answer here since one needs more data, but a broader discussion of *why* you expect differences and what they're driven by is needed here I think.

Nitpicks

=====

1. line 206 'privileges some information sources of others in an ad-hoc manner' -- possible typo, of=over?

2. In the supp' using the phrase "so called" in the reference to "so called mutual exclusivity" seems unnecessary. I get that you don't really buy this as an explanation on its own but I'm not sure there's a need for this.

3. The use of (i, ii, iii, iv) in Figure 1 is a tiny bit confusing, the first one is used _before_ the relevant sentence and the others (ii,iii,iv) are used after the relevant sentence

Reviewer #3:

Remarks to the Author:

Key results

This work provides both theoretical and methodological contributions that I believe are quite significant. Theoretically, it provides a quantitatively-specified developmental theory of how children integrate different information sources when learning their vocabulary, situated in a process of recursive social reasoning. A key finding is that the integration process itself does not qualitatively change between the ages of two and five --- children make rational use of their view of the information in the available cues.

Methodologically, this work uses formal computational cognitive models to implement that developmental theory, and evaluate its predictions with behavioral data collected from children. This quantitative framework demonstrates how developmental theories can be specified, implemented, and evaluated concretely.

Another impressive aspect is the careful consideration and evaluation of alternative models that consider qualitative developmental change in integration (the biased model and the developmental bias model). A comparison between those models and the rational integration model highlights that the rational integration model is indeed the best of these options at explaining children's behavior;

this means there's no qualitative change in cue integration at the ages, and so there's developmental continuity. To me, this part is a really valuable contribution because it separates out the information sources vs. the integration process, in terms of what's developing in children. That is, it's not just that the rational integration model could explain children's behavior well (i.e., an existence proof) -- this rational integration model did better than other integration models concretely implemented and using the same information sources as the rational integration model. So, really, we have stronger support for the developmental theory that says the integration process doesn't change (and that it's rational).

****Originality and significance****

In addition to the key ideas and contributions noted above, I really liked this approach to parameterizing the child's imagined lexicon for the literal listener. I agree that θ_{ij} then provides a more intuitive interpretation of word familiarity for word j that can be linked to empirical estimates of degree of acquisition at a particular age i . This is a very valuable feature of this approach.

Also, I think it's a major contribution to use RSA for developmental theorizing; I'm aware of developmental work that has been done with RSA before (e.g., Savinelli et al. 2017; Savinelli, K. J., Scontras, G., & Pearl, L. (2017). Modeling scope ambiguity resolution as pragmatic inference: Formalizing differences in child and adult behavior. In CogSci; Scontras & Pearl 2020 under review: When pragmatics matters more for truth-value judgments: An investigation of quantifier scope ambiguity.).

But, I haven't seen this comprehensive of an approach to both evaluation and alternative developmental hypotheses.

****Data & methodology****

As mentioned above, I'm very in favor of the way developmental data were used in combination with the modeling methodology. One aspect of the modeling I thought in particular was very well done was using the first two experiments to estimate model parameter values, and the third experiment to evaluate the model with those set values. This is akin to the separation of training vs. test data in NLP approaches, which is very good practice for preventing model overfitting (that can then hinder model generalization).

****Suggested improvements****

I had a few minor comments:

(1) Is there any utility in talking about RSA's α as a contrast parameter? That is, $\alpha > 1$ = turn up the relative contrasts between different probabilities vs. $\alpha < 1$ = smooth out the relative contrasts between different probabilities. (This is a way that makes a lot of intuitive sense to me, and I've been thinking about it this way for my own RSA models, but maybe it's not as helpful for the broad audience intuitions here). I could imagine children having a higher α (preferring sharper contrasts) than adults (i.e., making things more categorical than they actually are). I don't think we have adult data to compare against, but perhaps we might see this kind of change in α in children from ages 2 to 5. If so, then that may be useful to point out and link to prior work -- like that of Hudson Kam & Newport -- which shows children making probabilities more categorical than the input

warrants in the presence of inconsistent input.

(2) A related point to the one above: In the supplementary materials, I see the overall values for the three parameters (slope and intercept), but I assume these are aggregated across ages 2 to 5 since you don't show values for 2- vs. 3- vs. 4-year-olds. But I think you would have used different values for 2- vs. 3- vs. 4-year-olds? If so, it would be great to see those in the supplementary materials. Otherwise, I think I may have misunderstood how you set the parameter values from Experiments 1 and 2, and then used them to predict child behavior from 2 to 5 in Experiment 3.

(3) I like the point you're aiming to make about the three variables happening at different timespans, but is it fair to say that speaker informativeness is momentary and only active at the utterance level? It seems like this also might be based on the overall discourse or even longer-term knowledge about this speaker (or speakers in general).

(4) Related to the idea that a novel contribution is qualitative similarity in cue integration between ages 2 and 5, perhaps it would be helpful to highlight this in the discussion. That is, the key findings is that parameter values are changing in the rational integration model, but that the rational integration process is still the best fit. So, this is qualitative similarity across the ages, representing developmental continuity in how children are using the information they perceive.

(5) It's hard to manipulate speaker informativeness directly, I imagine, though it would have been great to manage this and so test the third parameter (α) in the same way that the other two were.

(6) For the biased integration model: It may be helpful to remind readers why the bias value ϕ can't have the same training vs. test set split that the rational integration model originally used. (This is because Experiment 3 is a combination of mutual exclusivity (Experiment 1) and speaker informativeness (Experiment 2), so the only way to identify their weighted combination is to use data from Experiment 3.) I had to revisit the experimental setup description to remind myself why.

(7) Related small clarity point: The authors might note in the text that the rational integration model trained on Expt 1-3 now explains 87% of the variance (this is better than the 79% from before). This appears in Figure 3 already, but not in the text walking through the comparative results.

(8) Another small clarity point: I found figure 1 to be really lovely and informative. But, as I was following along in the caption, I noticed that it might be helpful when you mention the developmental trajectories in (c) - (e) to mention the specific information sources at the point (i.e., say that these are developmental trajectories of common ground, speaker informativeness, and semantic knowledge, which are then pointed out in the experimental setup in (a) and the equations in (f)).

****References****

I think the previous literature cited looks quite sensible. I did wonder about related cue integration approaches though, such as the one by Gagliardi, Feldman, & Lidz 2017.

Gagliardi, A., Feldman, N. H., & Lidz, J. (2017). Modeling statistical insensitivity: Sources of suboptimal behavior. *Cognitive science*, 41(1), 188-217.

There, comparison models look at broken integration in two different ways, both of which seem to be a bit different than the biased integration and developmental bias approach pursued here. In both of those ways, Bayesian inference is still assumed, but the equivalent of bias is implemented as ignoring information from cues in different ways. (Gagliardi et al. 2017 also look at broken representations of cue information in one way as well.)

****Clarity and context****

(1) I think the opening of the introduction nicely situates the specific problem of word-learning for a broad audience by first highlighting why language is interesting, and then delving into why word learning in particular is interesting.

(2) The introduction also clearly spells out the key contributions of (i) implementing the proposed theory (that word-form learning involves social inference) in a computational cognitive model so that it can make precise, quantitative predictions, and (ii) comparing the implemented theory against alternative theories implemented in models that make different theoretical assumptions about whether information is included, how it's included, and how the inference process develops.

~ ~ ~

I like to sign my reviews when possible. ~ ~ Lisa S. Pearl

Author Rebuttal to Initial comments

Dear Dr. Antusch,

Thank you very much for considering our manuscript. Your comments and those of the reviewers have been very helpful for revising our manuscript. Below we include a detailed point-by-point response to all the comments made by the reviewers, to which we also responded with changes in the text (highlighted in red). More specifically, we have worked in the literature suggestions made by Reviewer 1 and clarified the points Reviewer 2 and 3 raised about our model.

Independent of the reviewers comments, we re-visited parts of our statistical analysis. We decided to re-run the Bayes Factor computations in a way that allowed us to use the same parameter estimates for all models. We believe that this strengthens the model comparison. As a consequence, the Bayes Factors differ numerically from the initial submission; the overall pattern of results remains the same.

Thank you very much for considering our manuscript and please do not hesitate to contact us with questions or concerns.

Sincerely,

Manuel Bohn

Reviewer #1:

1. The authors propose a model to account for children's integration of social-pragmatic information in word learning, drawing upon a Bayesian framework. I was quite surprised that no reference was made in the paper to Xu & Tenenbaum's 2007 model in Psychological Review (i.e., Word learning as Bayesian inference). I recognize that the model proposed in that paper is capturing a different aspect of word learning but it seems like it deserves some attention here.

Response: Thank you for the reference; Xu & Tenenbaum 2007 has been very influential for our group, and this was purely an issue of space. We added a reference to Xu & Tenenbaum (and other previous modelling work) to the introduction on page 4 and briefly point out what we take from their work and how our approach differs from theirs. The main similarity is the use of Bayesian inference, while the principal difference is that we look at word learning as the outcome of a social inference process instead of a cross-situational or principle-based learning process.

2. In a similar vein, I found the set up for the model somewhat disconnected. That is, the model is accounting for children's integration of two particular informational sources. This is not to say that this is not important but the abstract and introduction lead the reader to expect that a wide variety of word learning conditions are tested (e.g., "a range of experimental conditions"). It would be helpful to state up front the two informational sources that are being tested.

Response: Thanks for this feedback, we agree it would be helpful to readers to have a better sense of what is coming next. We changed the abstract to make it clear that a "range of experimental conditions" refers to combinations of three specific information sources.

3. Within the language processing literature, a great deal of attention has been paid to questions around integration of multiple constraints including the proposal of Bayesian models –the Heller et al model in particular (which is cited in the paper). I think it would be helpful to indicate how this proposed model connects with and extends this literature.

Response: We amended the discussion on page 13 to highlight how our work differs from earlier work on information integration in language processing. While previous work has focused on integrating perceptual, semantic or syntactic information, we extend the study of information integration to pragmatics. We also discuss the shared assumptions underlying our model and the model by Heller and colleagues on page 15 (i.e. that common ground changes how likely an object is to be the referent of an utterance).

4. The introduction notes that studies have typically examined children's attention to information in isolation. Other theorists have sought to provide more integrative accounts of word learning (e.g., Golinkoff et al., etc...). I think some acknowledgement of this literature is relevant here.

Response: Thanks, this framing was a bit overbroad; our intent was to highlight a gap in computational and formal work on integration. We discuss earlier empirical and theoretical work on information integration in the introduction on page 3 (now including a reference to Golinkoff's work). We think our main contribution in light of this earlier work is that we specify the nature of this integration process - which allows us to quantitatively predict children's learning in new experimental conditions.

Furthermore, other studies have sought to examine children's attention to convergent and divergent cues, with particular focus on the mutual exclusivity type phenomenon that is part of the focus in this model. I note some of these papers below. Would the model account for the findings from these types of models?

Gangopadhyay, I., & Kaushanskaya, M. (2020). The role of speaker eye gaze and mutual exclusivity in novel word learning by monolingual and bilingual children. *Journal of Experimental Child Psychology*, 197, 104878.

Jaswal, V. K., & Hansen, M. B. (2006). Learning words: Children disregard some pragmatic information that conflicts with mutual exclusivity. *Developmental science*, 9(2), 158-165.

Jaswal, V. K., & Markman, E. M. (2003). The relative strengths of indirect and direct word learning. *Developmental Psychology*, 39(4), 745.

Jaswal, V. K. (2010). Explaining the disambiguation effect: Don't exclude mutual exclusivity. *Journal of Child Language*, 37(1), 95-113.

Graham, S. A., Nilsen, E. S., Collins, S., & Olineck, K. (2010). The role of gaze direction and mutual exclusivity in guiding 24-month-olds' word mappings. *British Journal of Developmental Psychology*, 28(2), 449-465.

Paulus, M., & Fikkert, P. (2014). Conflicting social cues: Fourteen- and 24-month-old infants' reliance on gaze and pointing cues in word learning. *Journal of Cognition and Development*, 15(1), 43-59.

Response:

Thank you very much for pointing out these connections. The papers listed above (see also Grassmann & Tomasello, 2010) investigate how additional information sources (mostly social cues like eye-gaze or gestures) modify the mutual exclusivity inference. We think that our modelling framework would be very well suited to study how this additional information would be integrated. This is because we assume that all information sources flow together in a unified social inference process. Social information like

eye-gaze or gestures would be part of the utterance and thus affect the likelihood term in our model (see Bohn & Frank, 2019 for a proposal in this direction). Furthermore, in previous work (Bohn, Tessler, Merrick & Frank, 2019), we have used a structurally similar model to study how pointing gestures (non-verbal utterances) are integrated with common ground. In the discussion on page 14, we now point to this literature and suggest that it would be a very interesting route to test and extend our model further.

5. One core principle in the model is the way children integrate information sources remains constant. This is an interesting proposal and I appreciate that the model seems to capture this. I wonder, however, how this aligns with other research demonstrating increases in integrative capacity during the preschool years (i.e., changes in executive function). Can the authors expand on this proposal and align it with research demonstrating changes in EF?

Response: This is a very interesting point, thank you for bringing it up. The EF construct does include some elements of set-shifting (e.g., leveraged by the dimensional change card sort paradigm), but it is unknown whether this modulation of information sources via explicit rules is related to the kind of integration processes we study here. This question has been of persistent interest to us. One of our earlier studies (Horowitz*, Schneider* et al, 2017, Child Dev) used an individual differences paradigm to look at EF of this sort and its relation to pragmatic abilities and did not find any relations. In addition, we are currently investigating how executive functions relate to other different forms or pragmatic inference, including mutual exclusivity. However, we have not yet started modelling this interplay in any way.

Our - very preliminary - take on this would be that increases in EF might facilitate access to and maintenance of the different information sources – invoking other aspects of the EF construct that relate more to working memory rather than set shifting. On this view, increases in EF wouldn't necessarily change the way that information is integrated, consistent with our findings here. However, this is just a guess at the moment and we have no work to back this up. Thus, we would prefer to not discuss this topic in the paper because we feel we could only speculate about it.

6. No justification for sample sizes are provided. Notably the sample sizes in Experiment 3 are much larger than the sample sizes in previous experiments. It would be helpful to provide some justification for this decision.

Response: Thanks for flagging this issue; our preregistered sample sizes here were indeed larger than the standard in this literature in order to increase our quantitative precision. The divergence in sample sizes is due to the larger number of conditions in Experiment 3 compared to the other two. In the pre-registration, we noted that, in Experiment 3, we aimed for 30 data points per cell (with a cell being each combination of condition, item and age-group). This number was based on Experiment 1 and previous studies with similar tasks in which we found that 30 data points gives us precise enough estimates to be able to differentiate between conditions. We now include this information in the methods section on page 17.

Minor issues:

p. 3: It is noted that “referents can only be identified inferentially by reasoning about a speaker’s intentions”. This statement is not quite accurate as children can, of course, identify word referents through other means (tracking cross-situational occurrences etc...).

Response: Thanks for bringing this up. We agree that hypotheses about word meaning can be identified cross-situationally, but we’d argue that reference specifically is a construct about speakers’ intentions. (This distinction gets misused a bit in some of the early literature on this topic, including some of our own early work!). In addition, in naturalistic contexts, cross-situational word-learning is also mediated by reasoning about speakers’ intentions because there is typically no other unique cue to reference (see Frank, Tenenbaum & Fernald, 2013). Frank, Goodman & Tenenbaum (2009) present a model in which reference resolution in the moment is a process of social reasoning, even as word meaning accrues through cross-situational learning. Thus, we would feel comfortable to keep this claim, though we can also revisit it if the reviewer feels strongly.

Line 105: word-learning experiment (not world-learning)

Response: Thank you for spotting this - we corrected it accordingly.

Reviewer #2:

It's the nature of positive reviews that they are shorter than less-positive reviews so I don't have much to add. Please see the comments below for some suggestions, but I don't see these are barriers for publication.

Thank you very much for your kind words!

*1. In lines 40-60, one could wrongly get the impression that the reality is that children use a "bag of tricks", and the question is now how to model it. That is, something like "Previous work has considered A, B, C in isolation but how do they work together? What're the weights a,b,c, such that JUDGMENTS = a*A + b*B + c*C?". I think you could add another few lines somewhere around line 60 to more clearly state your integration theory ISN'T a bag-of-tricks view and that you will test it against such one.*

Response: Thank you for pointing this out, we appreciate you clarifying this aspect of the prose. We now emphasize the difference between our model and the alternatives we consider (including the bag of tricks) in the introduction on page 5.

2. I understand 'sensitivity to informativeness' (alpha) is defined based on previous work but I still don't quite get from the opening how this is supposed to work. I understand a parameter relating to "do I know this word" as pretty directly grounded in some mental variable, I understand a parameter relating to common ground as "Did I notice you weren't here" (maybe), and their change over time could reasonably

reflect development of competence, memory, knowledge, etc. What exactly is growing sensitivity to informativeness meant to represent? Do we think the child is able to think of the speaker as informative, but what they grow to understand is that this...matters? It seems unlikely, and probably I'm missing an obvious point here, but it seems like sensitivity to informativeness is likely tracking a bunch of more complicated processes having to do with whether the child themselves can reason about the speaker, can calculate informativeness, etc. As such, this parameter seems more like it is soaking up noise/growing competence than a specific thing like 'memory for this object'.(see also comment 5 on the interpretation of 'sensitivity to common ground').

Response: Thanks for the interesting question. First, clarifying from a mathematical perspective, the exponent alpha is a nice way of putting a parameter on how informative speakers are. If $\alpha = 0$, then speakers are assumed by the learner to choose between true words randomly, while when alpha goes to infinity, speakers always choose the most informative word. So this parameter lets us infer the degree of informativeness as a continuous quantity rather than a discrete switch (“is the speaker Gricean or no”).

If one were to give a psychological interpretation of alpha, it could be as follows: In the model, we assume that children make their inference based on the assumption that the speaker is trying to be helpful or cooperative (in a Gricean sense) by choosing an informative word. In our case, helpful means basically how carefully they choose their words relative to what they mean to convey. So an inferred value of alpha then tells you how helpful in that sense the child thinks the speaker is.

However, we would like to emphasize that our model is first and foremost a computational description of the inference and we therefore make no strong claims about the psychological reality of the processes or parameters in it. In fact, we generally agree with the reviewer that this parameter is quite a high level description of what is likely a complex set of inferences. We now acknowledge this point on page 16 in the discussion.

3. Following on (2), if I understand correctly your experiments do not independently verify the parameters for informativeness and semantic knowledge, rather those are both being estimated at the same time from Exp 1. Is there a reason not to do this independently? Is it simply not possible?

Response: That’s right, Experiment 1 jointly estimates the parameters for informativeness and semantic knowledge. In the revised paper, we make this more clear in the results section. In theory, it would be possible to estimate these parameters independently. We would need two different experiments, a vocabulary test for the objects in the study and an experimental task measuring informativeness that is suited for children between 2 and 5 and that could be modeled using a structurally similar model compared to the one we are using here. We are currently working on such an additional task, but, as of now, it is unclear if it would be suitable to independently estimate this parameter. Furthermore, because our focus was on information integration, we tried to avoid different experiential stimuli in Exp. 1, 2 and Exp. 3. We totally agree that this is an interesting way to follow-up on the work presented here.

4. *Looking at the supp', Figure S6: What's going on with some of the cases where the prediction seems to be going in the opposite direction than the data, like horseshoe, plug, barrel, pawn? Is this just chalked up to noise or is there something more principled here?*

Response: The developmental trajectory for the data in Figure S6 is visualized using a simple GLM smoother, which is independently estimated for each cell. This model is rather simple and less flexible compared to our cognitive model. Thus, we don't think that the curve is a particularly good representation of the "real" developmental trajectory of the data. We added it because it gives the reader an idea about the general region in which the average rate of responses lies in comparison to what the model predicts. But, we don't think these differences in direction are particularly meaningful.

5. *(around lines 312-314) "But computationally, the model does not differentiate between common ground information and other reasons that might make an object contextually more salient..." -- Right, I had this thought earlier in the manuscript. Is there any way to tease these apart? It also makes a lot more sense to me personally to call the relevant parameter (currently 'sensitivity to common ground') something that refers to this attention-grabbing aspect of the object, whether it is novelty, shininess, blinking, etc. Calling it 'sensitivity to common ground' seems overly specific, and like 'sensitivity to informativeness' seems unlikely to refer to a natural kind. This should probably be discussed/acknowledged much higher up when introducing the 'common ground' bits of the model.*

Response: Thank you for this suggestion. In the revised paper, we now make it clear when we introduce the model, that treating the prior as representing common ground is something that follows from the experiments we used and not from the way that we modeled it (see page 5 and 15).

6. *Cultural differences (around lines 299-301): The authors acknowledge their results will likely not replicate in a strict sense, which is fair enough. But...what exactly do you think will be driving this cultural variation? If the underlying process is the same and the integration is the same, this leaves us with children having different knowledge of objects in different cultures (which is fair enough, perhaps in some cultures they know about more/less objects, perhaps not _these objects), or having 'different sensitivity to common ground / informativeness'. The latter is much weirder and you'd need to explain even briefly how that would be driven by a cultural difference. I'm not expecting an answer here since one needs more data, but a broader discussion of *why* you expect differences and what they're driven by is needed here I think.*

Response: The reason we expect variation in sensitivity to information sources is that studies looking at pragmatic inferences across cultures and languages (similar to the ones we are investigating here) find such variation. Where this variation comes from (i.e. which cultural practises influence these inferences we don't have the data to tell. In the revised paper on page 15, we now explain why we expect there to be differences across cultures and refer to the respective literature.

Nitpicks

=====

1. line 206 'privileges some information sources of others in an ad-hoc manner' -- possible typo, of=over?

Response: Fixed it, thank you.

2. In the supp' using the phrase "so called" in the reference to "so called mutual exclusivity" seems unnecessary. I get that you don't really buy this as an explanation on its own but I'm not sure there's a need for this.

Response: Thank you, we changed the phrasing.

3. The use of (i, ii, iii, iv) in Figure 1 is a tiny bit confusing, the first one is used before the relevant sentence and the others (ii,iii,iv) are used after the relevant sentence

Response: We now use a consistent ordering.

Reviewer #3:

This work provides both theoretical and methodological contributions that I believe are quite significant. Theoretically, it provides a quantitatively-specified developmental theory of how children integrate different information sources when learning their vocabulary, situated in a process of recursive social reasoning. A key finding is that the integration process itself does not qualitatively change between the ages of two and five --- children make rational use of their view of the information in the available cues.

Methodologically, this work uses formal computational cognitive models to implement that developmental theory, and evaluate its predictions with behavioral data collected from children. This quantitative framework demonstrates how developmental theories can be specified, implemented, and evaluated concretely.

Another impressive aspect is the careful consideration and evaluation of alternative models that consider qualitative developmental change in integration (the biased model and the developmental bias model). A comparison between those models and the rational integration model highlights that the rational integration model is indeed the best of these options at explaining children's behavior; this means there's no qualitative change in cue integration at the ages, and so there's developmental continuity. To me, this part is a really valuable contribution because it separates out the information sources vs. the integration process, in terms of what's developing in children. That is, it's not just that the rational integration model could explain children's behavior well (i.e., an existence proof) -- this rational integration model did

better than other integration models concretely implemented and using the same information sources as the rational integration model. So, really, we have stronger support for the developmental theory that says the integration process doesn't change (and that it's rational).

Response: Thanks so much for the kind words!

In addition to the key ideas and contributions noted above, I really liked this approach to parameterizing the child's imagined lexicon for the literal listener. I agree that θ_{ij} then provides a more intuitive interpretation of word familiarity for word j that can be linked to empirical estimates of degree of acquisition at a particular age i . This is a very valuable feature of this approach.

Also, I think it's a major contribution to use RSA for developmental theorizing; I'm aware of developmental work that has been done with RSA before (e.g., Savinelli et al. 2017: Savinelli, K. J., Scontras, G., & Pearl, L. (2017). Modeling scope ambiguity resolution as pragmatic inference: Formalizing differences in child and adult behavior. In CogSci; Scontras & Pearl 2020 under review: When pragmatics matters more for truth-value judgments: An investigation of quantifier scope ambiguity.).

But, I haven't seen this comprehensive of an approach to both evaluation and alternative developmental hypotheses.

Response: Thank you for pointing us to this work, we now include it in the manuscript.

As mentioned above, I'm very in favor of the way developmental data were used in combination with the modeling methodology. One aspect of the modeling I thought in particular was very well done was using the first two experiments to estimate model parameter values, and the third experiment to evaluate the model with those set values. This is akin to the separation of training vs. test data in NLP approaches, which is very good practice for preventing model overfitting (that can then hinder model generalization).

Response: Thank you again, we very much appreciate this comment.

I had a few minor comments:

(1) Is there any utility in talking about RSA's alpha as a contrast parameter? That is, $\alpha > 1$ = turn up the relative contrasts between different probabilities vs. $\alpha < 1$ = smooth out the relative contrasts between different probabilities. (This is a way that makes a lot of intuitive sense to me, and I've been thinking about it this way for my own RSA models, but maybe it's not as helpful for the broad audience intuitions here). I could imagine children having a higher alpha (preferring sharper contrasts) than adults (i.e., making things more categorical than they actually are). I don't think we have adult data to compare against, but perhaps we might see this kind of change in alpha in children from ages 2 to 5. If so, then that may be useful to point out and link to prior work -- like that of Hudson Kam & Newport --

which shows children making probabilities more categorical than the input warrants in the presence of inconsistent input.

Response: That's an interesting way to think about alpha in this context. However, in our case, all alpha values were above 1. So, it only served as an amplifier. What we also see in our data (e.g. Figure 1d) is a steady increase in alpha with age. In a different study - where we used a similar model but with data from both adults and children - we also found higher values for alpha in adults compared to children. But this interpretation of alpha might, to some extent, depend on the way our model is structured and might be different for other models.

(2) A related point to the one above: In the supplementary materials, I see the overall values for the three parameters (slope and intercept), but I assume these are aggregated across ages 2 to 5 since you don't show values for 2- vs. 3- vs. 4-year-olds. But I think you would have used different values for 2- vs. 3- vs. 4-year-olds? If so, it would be great to see those in the supplementary materials. Otherwise, I think I may have misunderstood how you set the parameter values from Experiments 1 and 2, and then used them to predict child behavior from 2 to 5 in Experiment 3.

Response: We sampled continuously between age 2 and 5 and only binned the data for the correlational analysis. Thus, we estimated a single slope and intercept for each information source based on the data from all age groups together. This combination of slope and intercept describes the line that represents the average responses of children for each information source across the entire age range. To then compute the parameter values for a particular age, we used the intercept and added the slope times the age (e.g. $\alpha_i = b_{\alpha_intercept} + b_{\alpha_slope} * age_i$). This way we could compute different parameter values not just for each age group (2-, 3-, and 4-year-olds) but for any age between 2 and 5 (e.g. 3.45). These values were then fed into the model to generate model predictions for that particular age. The upper right legend of Figure S4 spells out these computations. We've also changed the text in the supplementary material (section: Loci of development) to hopefully make this more clear.

(3) I like the point you're aiming to make about the three variables happening at different timespans, but is it fair to say that speaker informativeness is momentary and only active at the utterance level? It seems like this also might be based on the overall discourse or even longer-term knowledge about this speaker (or speakers in general).

Response: That is a really interesting idea, thank you. It sounds a bit like the kind of adaptation described in Schuster & Degen (2020), for example. In our work, computationally, the informativeness assumption is built into the model structure (reflecting the Gricean principle of cooperative communication) and is only modified by the parameter alpha. As such it is always an assumption about a particular set of utterances in a particular context. In our case, the children had no a-priori information about the different speakers (they were all different animal characters). But it's conceivable that, over time, children build up expectations that are more specific to the identity of the speaker. Depending on the kinds of experience,

this could result in a speaker specific expectation about how informative this speaker is. That is a really interesting idea to follow up in the future.

(4) Related to the idea that a novel contribution is qualitative similarity in cue integration between ages 2 and 5, perhaps it would be helpful to highlight this in the discussion. That is, the key findings is that parameter values are changing in the rational integration model, but that the rational integration process is still the best fit. So, this is qualitative similarity across the ages, representing developmental continuity in how children are using the information they perceive.

Response: Great thought. We emphasize this point in the revised paper on page 14.

(5) It's hard to manipulate speaker informativeness directly, I imagine, though it would have been great to manage this and so test the third parameter (alpha) in the same way that the other two were.

Response: We agree, that is a really interesting way to extend our work. We added this to the paper on page 16.

(6) For the biased integration model: It may be helpful to remind readers why the bias value phi can't have the same training vs. test set split that the rational integration model originally used. (This is because Experiment 3 is a combination of mutual exclusivity (Experiment 1) and speaker informativeness (Experiment 2), so the only way to identify their weighted combination is to use data from Experiment 3.) I had to revisit the experimental setup description to remind myself why.

Response: We re-phrased the section describing this situation on page 11 to make this more clear.

(7) Related small clarity point: The authors might note in the text that the rational integration model trained on Expt 1-3 now explains 87% of the variance (this is better than the 79% from before). This appears in Figure 3 already, but not in the text walking through the comparative results.

Response: We added this information on page 11.

(8) Another small clarity point: I found figure 1 to be really lovely and informative. But, as I was following along in the caption, I noticed that it might be helpful when you mention the developmental trajectories in (c) - (e) to mention the specific information sources at the point (i.e., say that these are developmental trajectories of common ground, speaker informativeness, and semantic knowledge, which are then pointed out in the experimental setup in (a) and the equations in (f)).

Response: We changed the caption accordingly.

****References****

I think the previous literature cited looks quite sensible. I did wonder about related cue integration approaches though, such as the one by Gagliardi, Feldman, & Lidz 2017.

Gagliardi, A., Feldman, N. H., & Lidz, J. (2017). Modeling statistical insensitivity: Sources of suboptimal behavior. Cognitive science, 41(1), 188-217.

There, comparison models look at broken integration in two different ways, both of which seem to be a bit different than the biased integration and developmental bias approach pursued here. In both of those ways, Bayesian inference is still assumed, but the equivalent of bias is implemented as ignoring information from cues in different ways. (Gagliardi et al. 2017 also look at broken representations of cue information in one way as well.)

Response: Thank you for pointing us to this paper, we now include it as an example that children might sometimes ignore information sources during language learning on page 8.

Clarity and context

(1) I think the opening of the introduction nicely situates the specific problem of word-learning for a broad audience by first highlighting why language is interesting, and then delving into why word learning in particular is interesting.

(2) The introduction also clearly spells out the key contributions of (i) implementing the proposed theory (that word-form learning involves social inference) in a computational cognitive model so that it can make precise, quantitative predictions, and (ii) comparing the implemented theory against alternative theories implemented in models that make different theoretical assumptions about whether information is included, how it's included, and how the inference process develops.

Response: We are glad our framing was helpful, thank you.

Decision Letter, first revision:

Our ref: NATHUMBEHAV-201013052A

14th April 2021

Dear Dr. Bohn,

Thank you for submitting your revised manuscript "How young children integrate information sources to infer the meaning of words" (NATHUMBEHAV-201013052A). It has now been seen by the original referees and their comments are below. As you can see, the reviewers find that the paper has improved in revision. We will therefore be happy in principle to publish it in Nature Human Behaviour, pending minor revisions to satisfy the referees' final requests and to comply with our editorial and formatting guidelines.

Reviewer 1 signed off without further comments. Reviewer 2 had minor suggestions which we ask you to consider when revising your manuscript. Finally, Reviewer 3 raised an important point about the psychological interpretation of the alpha parameter included in your model and the use of it to explain real-world information integration in children. We agree with the reviewer and ask you to more clearly acknowledge this limitation of your model in the manuscript and emphasize that you make no strong claims about the psychological reality of the processes or parameters in it.

Sincerely,

Samantha Antusch
Editor
Nature Human Behaviour

Reviewer #1 (Remarks to the Author):

I thank the authors for the attention to the issues raised in the previous round of reviews. I have no further comments.

Reviewer #2 (Remarks to the Author):

I like to sign my reviews when possible. ~ ~Lisa S. Pearl

I continue to believe this work provides both theoretical and methodological contributions that are quite significant, for all the reasons I noted in my prior review. I think the revisions that the authors have made in response to my very few suggestions from last time are very helpful, and have clarified minor points of potential confusion that may have arisen. I have some other very minor things I might suggest updating in this version (detailed below), and I very much support publication of this highly interesting and impactful work.

Very minor things:

(1) In the introduction, I'm not sure it's helpful to contrast your approach against Xu & Tenenbaum's (2007) approach by saying theirs is a principle-based learning process. That makes it sound like yours isn't, and I very much think it is. Perhaps a better framing is that yours incorporates a principle-based learning process (in fact, it's the same principle as Xu & Tenenbaum's: Bayesian inference) as part of a larger social inference process.

(2) In the results section, the callout to the Gagliardi et al. 2017 work fits well where you put it (highlighting that prior work shows children's behavior in other circumstances is best captured by

selectively ignoring certain information). I'd recommend a slight adjustment to how you characterize that work, though -- rather than being about word learning, I believe that work was about noun classification into morphology classes. So, maybe it's more accurate to talk about it as "another area of language development" or "morphology learning that involved semantic information" rather than word learning.

(3) This is a suggested wording tweak, because I'm not sure I followed the "On the contrary" segue otherwise on line 186-189 at the end of "Predicting information integration across development". Did you mean something like "In contrast with these lesioned models that underestimate performance in the incongruent condition"? If so, then the information that follows makes more sense (with the no-common-ground model overestimating performance).

Reviewer #3 (Remarks to the Author):

I was already pretty well-disposed towards the paper and I appreciate the authors taking the time to address my outstanding concerns. I'm happy to sign off on it.

But. In what follows I'm going to nitpick on one of the responses (specifically to point 2, to R2). This isn't a barrier for publication or anything, and if the authors leave it as currently is that's fine too, but since it needles me:

In discussing the 'alpha' parameter, I asked for an interpretation of what it means or could mean, as a psychological variable, and I worried that it seems odd in psychological-plausibility relative to the other parameter. The authors explained what it means from the modeling perspective (which I get, and wasn't the issue to begin with), and then offer the following:

"If one were to give a psychological interpretation of alpha, it could be as follows: In the model, we assume that children make their inference based on the assumption that the speaker is trying to be helpful or cooperative (in a Gricean sense) by choosing an informative word. In our case, helpful means basically how carefully they choose their words relative to what they mean to convey. So an inferred value of alpha then tells you how helpful in that sense the child thinks the speaker is."

To me this is just bringing the weirdness of this parameter to the foreground. Phrased this way, it asks us to believe children can at all points in development in principle calculate what a maximally informative answer *would* be (and all other levels of informativeness) but that in addition to this feat, they are placing some probability on the speaker actually deciding to be helpful in providing such information.

The learning process showing an increase in alpha then asks us to believe that children start out thinking "maybe people are unhelpful/uninformative", but over time children learn "oh you know what, I thought people are uninformative, but turns out they are"? This is a stretch. It seems highly implausible that children know from the outset what the maximally informative word is, and rather more plausible that changes to this parameter reflect a growing understanding of how to even calculate informativeness (again, as opposed to expectations that a speaker *will* be informative for a fixed understanding of informativeness).

Again, I get that this is a computational model without claims to a 1-to-1 mapping between parameters and reality, and I get that models specifically like this have already been out there in the literature, it's just that the other parameters *do* have more plausible connections to psychological reality and as phrased this one was the odd duck out. The clarification in the reviewer response doesn't really clear it up, though I would agree with the next part the authors say:

"However, we would like to emphasize that our model is first and foremost a computational description of the inference and we therefore make no strong claims about the psychological reality of the processes or parameters in it. In fact, we generally agree with the reviewer that this parameter is quite a high level description of what is likely a complex set of inferences. We now acknowledge this point on page 16 in the discussion."

Decision letter, final requests:

** Please ensure you delete the link to your author homepage in this e-mail if you wish to forward it to your co-authors. **

Our ref: NATHUMBEHAV-201013052A

5th May 2021

Dear Dr. Bohn,

Thank you for your patience as we've prepared the guidelines for final submission of your Nature Human Behaviour manuscript, "How young children integrate information sources to infer the meaning of words" (NATHUMBEHAV-201013052A). Please carefully follow the step-by-step instructions provided in the attached file, and add a response in each row of the table to indicate the changes that you have made. Ensuring that each point is addressed will help to ensure that your revised manuscript can be swiftly handed over to our production team.

Nature Human Behaviour offers a Transparent Peer Review option for new original research manuscripts submitted after December 1st, 2019. As part of this initiative, we encourage our authors to support increased transparency into the peer review process by agreeing to have the reviewer comments, author rebuttal letters, and editorial decision letters published as a Supplementary item. When you submit your final files please clearly state in your cover letter whether or not you would like to participate in this initiative. Please note that failure to state your preference will result in delays in accepting your manuscript for publication.

In recognition of the time and expertise our reviewers provide to Nature Human Behaviour's editorial process, we would like to formally acknowledge their contribution to the external peer review of your manuscript entitled "How young children integrate information sources to infer the meaning of words". For those reviewers who give their assent, we will be publishing their names alongside the published article.

Cover suggestions

As you prepare your final files we encourage you to consider whether you have any images or illustrations that may be appropriate for use on the cover of Nature Human Behaviour.

ORCID

Non-corresponding authors do not have to link their ORCIDs but are encouraged to do so. Please note that it will not be possible to add/modify ORCIDs at proof. Thus, please let your co-authors know that if they wish to have their ORCID added to the paper they must follow the procedure described in the following link prior to acceptance: <https://www.springernature.com/gp/researchers/orcid/orcid-for-nature-research>

Nature Human Behaviour has now transitioned to a unified Rights Collection system which will allow our Author Services team to quickly and easily collect the rights and permissions required to publish your work. Approximately 10 days after your paper is formally accepted, you will receive an email in providing you with a link to complete the grant of rights. If your paper is eligible for Open Access, our Author Services team will also be in touch regarding any additional information that may be required to arrange payment for your article. Please note that you will not receive your proofs until the publishing agreement has been received through our system.

Please note that *Nature Human Behaviour* is a Transformative Journal (TJ). Authors may publish their research with us through the traditional subscription access route or make their paper immediately open access through payment of an article-processing charge (APC). Authors will not be required to make a final decision about access to their article until it has been accepted. [Find out more about Transformative Journals](https://www.springernature.com/gp/open-research/transformative-journals)

Authors may need to take specific actions to achieve [compliance with funder and institutional open access mandates.](https://www.springernature.com/gp/open-research/funding/policy-compliance-faqs) For submissions from January 2021, if your research is supported by a funder that requires immediate open access (e.g. according to [Plan S principles](https://www.springernature.com/gp/open-research/plan-s-compliance)) then you should select the gold OA route, and we will direct you to the compliant route where possible. For authors selecting the subscription publication route our standard licensing terms will need to be accepted, including our [self-archiving policies](https://www.springernature.com/gp/open-research/policies/journal-policies). Those standard licensing terms will supersede any other terms that the author or any third party may assert apply to any version of the manuscript.

[REDACTED]

Best regards,
Chloe Knight
Editorial Assistant
Nature Human Behaviour

On behalf of

Samantha Antusch
Editor
Nature Human Behaviour

Reviewer #1:

Remarks to the Author:

I thank the authors for the attention to the issues raised in the previous round of reviews. I have no further comments.

Reviewer #3:

Remarks to the Author:

I like to sign my reviews when possible. ~ ~Lisa S. Pearl

I continue to believe this work provides both theoretical and methodological contributions that are quite significant, for all the reasons I noted in my prior review. I think the revisions that the authors have made in response to my very few suggestions from last time are very helpful, and have clarified minor points of potential confusion that may have arisen. I have some other very minor things I might suggest updating in this version (detailed below), and I very much support publication of this highly interesting and impactful work.

Very minor things:

(1) In the introduction, I'm not sure it's helpful to contrast your approach against Xu & Tenenbaum's (2007) approach by saying theirs is a principle-based learning process. That makes it sound like yours isn't, and I very much think it is. Perhaps a better framing is that yours incorporates a principle-based learning process (in fact, it's the same principle as Xu & Tenenbaum's: Bayesian inference) as part of a larger social inference process.

(2) In the results section, the callout to the Gagliardi et al. 2017 work fits well where you put it (highlighting that prior work shows children's behavior in other circumstances is best captured by selectively ignoring certain information). I'd recommend a slight adjustment to how you characterize that work, though -- rather than being about word learning, I believe that work was about noun classification into morphology classes. So, maybe it's more accurate to talk about it as "another area of language development" or "morphology learning that involved semantic information" rather than word learning.

(3) This is a suggested wording tweak, because I'm not sure I followed the "On the contrary" segue otherwise on line 186-189 at the end of "Predicting information integration across development". Did you mean something like "In contrast with these lesioned models that underestimate performance in the incongruent condition"? If so, then the information that follows makes more sense (with the no-common-ground model overestimating performance).

Reviewer #4:

Remarks to the Author:

I was already pretty well-disposed towards the paper and I appreciate the authors taking the time to address my outstanding concerns. I'm happy to sign off on it.

But. In what follows I'm going to nitpick on one of the responses (specifically to point 2, to R2). This isn't a barrier for publication or anything, and if the authors leave it as currently is that's fine too, but since it needles me:

In discussing the 'alpha' parameter, I asked for an interpretation of what it means or could mean, as a psychological variable, and I worried that it seems odd in psychological-plausibility relative to the other parameter. The authors explained what it means from the modeling perspective (which I get, and wasn't the issue to begin with), and then offer the following:

"If one were to give a psychological interpretation of alpha, it could be as follows: In the model, we assume that children make their inference based on the assumption that the speaker is trying to be helpful or cooperative (in a Gricean sense) by choosing an informative word. In our case, helpful

means basically how carefully they choose their words relative to what they mean to convey. So an inferred value of alpha then tells you how helpful in that sense the child thinks the speaker is."

To me this is just bringing the weirdness of this parameter to the foreground. Phrased this way, it asks us to believe children can at all points in development in principle calculate what a maximally informative answer *would* be (and all other levels of informativeness) but that in addition to this feat, they are placing some probability on the speaker actually deciding to be helpful in providing such information.

The learning process showing an increase in alpha then asks us to believe that children start out thinking "maybe people are unhelpful/uninformative", but over time children learn "oh you know what, I thought people are uninformative, but turns out they are"? This is a stretch. It seems highly implausible that children know from the outset what the maximally informative word is, and rather more plausible that changes to this parameter reflect a growing understanding of how to even calculate informativeness (again, as opposed to expectations that a speaker *will* be informative for a fixed understanding of informativeness).

Again, I get that this is a computational model without claims to a 1-to-1 mapping between parameters and reality, and I get that models specifically like this have already been out there in the literature, it's just that the other parameters *do* have more plausible connections to psychological reality and as phrased this one was the odd duck out. The clarification in the reviewer response doesn't really clear it up, though I would agree with the next part the authors say:

"However, we would like to emphasize that our model is first and foremost a computational description of the inference and we therefore make no strong claims about the psychological reality of the processes or parameters in it. In fact, we generally agree with the reviewer that this parameter is quite a high level description of what is likely a complex set of inferences. We now acknowledge this point on page 16 in the discussion."

Final Decision Letter:

Dear Dr Bohn,

We are pleased to inform you that your Article "How young children integrate information sources to infer the meaning of words", has now been accepted for publication in Nature Human Behaviour.

Before your manuscript is typeset, we will edit the text to ensure it is intelligible to our wide readership and conforms to house style. We look particularly carefully at the titles of all papers to ensure that they are relatively brief and understandable.

Acceptance of your manuscript is conditional on all authors' agreement with our publication policies

(see <http://www.nature.com/nathumbehav/info/gta>). In particular your manuscript must not be published elsewhere and there must be no announcement of the work to any media outlet until the publication date (the day on which it is uploaded onto our web site).

Please note that *Nature Human Behaviour* is a Transformative Journal (TJ). Authors may publish their research with us through the traditional subscription access route or make their paper immediately open access through payment of an article-processing charge (APC). Authors will not be required to make a final decision about access to their article until it has been accepted. [Find out more about Transformative Journals](https://www.springernature.com/gp/open-research/transformative-journals)

Authors may need to take specific actions to achieve compliance with funder and institutional open access mandates. For submissions from January 2021, if your research is supported by a funder that requires immediate open access (e.g. according to Plan S principles) then you should select the gold OA route, and we will direct you to the compliant route where possible. For authors selecting the subscription publication route our standard licensing terms will need to be accepted, including our self-archiving policies. Those standard licensing terms will supersede any other terms that the author or any third party may assert apply to any version of the manuscript.

To assist our authors in disseminating their research to the broader community, our SharedIt initiative provides you with a unique shareable link that will allow anyone (with or without a subscription) to

read the published article. Recipients of the link with a subscription will also be able to download and print the PDF.

With best regards,

Samantha Antusch
Editor
Nature Human Behaviour

P.S. Click on the following link if you would like to recommend Nature Human Behaviour to your librarian <http://www.nature.com/subscriptions/recommend.html#forms>

** Visit the Springer Nature Editorial and Publishing website at http://editorial-jobs.springernature.com?utm_source=ejp_NHumB_email&utm_medium=ejp_NHumB_email&utm_campaign=ejp_NHumB for more information about our career opportunities. If you have any questions please click [here](mailto:editorial.publishing.jobs@springernature.com). **